# Learning from Active Human Involvement through Proxy Value Propagation

**Zhenghao Peng**[§]**, Wenjie Mo**[§]**, Chenda Duan**[§]**, Quanyi Li**[†]**, Bolei Zhou**[§]
[§]University of California, Los Angeles, [†]University of Edinburgh

## Abstract

Learning from active human involvement enables the human subject to actively intervene and demonstrate to the AI agent during training. The interaction and corrective feedback from human brings safety and AI alignment to the learning process. In this work, we propose a new reward-free active human involvement method called *Proxy Value Propagation* for policy optimization. Our key insight is that a proxy value function can be designed to express human intents, wherein state-action pairs in the human demonstration are labeled with high values, while those agents' actions that are intervened receive low values. Through the TD-learning framework, labeled values of demonstrated state-action pairs are further propagated to other unlabeled data generated from agents' exploration. The proxy value function thus induces a policy that faithfully emulates human behaviors. Human-in-the-loop experiments show the generality and efficiency of our method. With minimal modification to existing reinforcement learning algorithms, our method can learn to solve continuous and discrete control tasks with various human control devices, including the challenging task of driving in Grand Theft Auto V. Demo video and code are available at: `https://metadriverse.github.io/pvp`.

## 1 Introduction

Reinforcement learning (RL) has been successfully applied in many domains, ranging from board game Go [46], strategy game StarCraft II [42], autonomous driving [18], and even nuclear fusion [7]. Existing RL methods assume the manually designed reward functions can fully express human intents and preferences. However, the resulting agents might exhibit biased, misguided, or undesired behaviors due to faulty reward functions [23, 40, 20]. Moreover, the poor sample efficiency as well as the safety concern due to the trial-and-error exploration prevent the real-world deployment of RL.

Human-in-the-loop methods are promising to achieve alignment, learning efficiency, and safety. Human-in-the-loop policy learning relies on human subjects to oversee the learning process of the autonomous agents, thus it can better align the learned behaviors with the preferences of humans compared with handcrafted reward functions. Different forms of human involvement in human-in-the-loop policy learning have been studied over the years. Human subjects can advise actions upon the requests of the robots [28] or provide preference-based feedback to assess the relative value of the collected trajectories [53, 5, 38, 52, 36, 11, 34]. These methods learn from passive human involvement, where the human subjects do not provide real-time feedback and intervention during data collection. For safety-critical tasks such as autonomous driving, safety is undoubtedly the first priority in human preference and the passive involvement methods yield unbounded risks in such settings. An increasing body of works focuses on active human involvement, where human subjects actively intervene and provide demonstrations during the execution time [17, 47, 29, 26]. With online correction and demonstration from human subjects, AI alignment and training-time safety of the system can be substantially enhanced.

37th Conference on Neural Information Processing Systems (NeurIPS 2023).

In this work, we focus on learning from active human involvement and develop a simple yet effective method that can turn a common value-based RL method into a reward-free human-in-the-loop method with minimal modification. Our key insight is that we can learn a proxy value function from active human involvement, such that the proxy values encode human intents and guide policy learning to emulate human behaviors. Specifically, we propose the *Proxy Value Propagation (PVP)* method which labels high Q values to human actions and low Q values to agent actions that are intervened by the human subjects. The proxy values are then propagated to unlabeled state-action pairs in the agent's exploration through TD-learning. Value-based RL methods soon learn policies that align with human intents because of the value-maximization nature. Experiments show that PVP can be successfully applied to both continuous and discrete action spaces, and achieve higher learning efficiency compared to baselines in various tasks, including driving in Grand Theft Auto V (GTA V). It is also compatible with different forms of human control devices, including gamepad, driving wheel, and keyboard. We summarize our main contributions as follows:

1) We propose a simple yet effective method, Proxy Value Propagation, that can be integrated into existing RL algorithms to learn from active human involvement. Our method is reward-free and can be generalized across various task settings and human control devices.

2) The experiments show that the proposed PVP method enables superior performance and high learning efficiency in various tasks from the MiniGrid, MetaDrive, CARLA, to GTA V environment. User study further shows that PVP achieves better performance and is more user-friendly compared to other human-in-the-loop baselines.

## 2   Related Work

AI alignment is one of the major issues in learning trustworthy intelligent agents for real-world applications. It is difficult to represent various human preferences into a scalar reward function in existing Reinforcement Learning (RL) methods [40, 6]. Meanwhile, the manually designed reward function, which might be misaligned with human preferences, often leads to undesired behaviors [23, 20]. As a promising complement to RL, Human-in-the-loop Learning (HL) can overcome costly reward engineering and convey human intents to the learning process directly through human involvement. Compared to imitation learning (IL) [14, 9], where the agent learns directly from high-quality human demonstration, HL methods benefit from interactive human involvement and feedback during the training, mitigating the possible distributional shift that usually happens when learning from offline data [39].

**Preference-based RL.** A large body of work focuses on learning human preference via ranking pair of trajectories generated by the learning agent [5, 11, 38, 52, 41, 36, 22, 51]. InstructGPT [34] aligns language models by first supervised learning in demonstration and then finetuning by the reward learned from human preference feedback. Preference learning can be applied to the tasks that human can not conduct, such as moving a six-legged Ant robot by assigning exact torque at each joint [5]. For those tasks that human can demonstrate, these methods do not fully utilize real-time feedback from human subjects during agent-environment interaction.

**HL with Passive Human Involvement.** Different from preference-based RL, human subjects can provide direct feedback to the learning agent during training through passive human involvement. Some works learn policy from human-provided evaluative feedback, a Boolean flagging correct or wrong actions [19, 3, 32]. This is similar to the intervention in our framework. However, in [32], humans provide high-level instructions, e.g. pointing to the left/right, while in PVP humans provide intervention and low-level demonstrations. The other line of work allows the neural policy to operate the robot and the human subjects can provide demonstration upon the requests from the learning agents [28, 30, 16]. The expert policy will intervene when uncertainty is huge, where the agent uncertainty is estimated by the variance of actions [30]. These methods reduce the cost of human resources but have potential risks to human subjects since they do not fully control the system. For example, when human subjects use these algorithms to train autopilot AI, they are exposed to significant risks if they are in a self-driving cars due to unpredictable agent behaviors.

**Learning from Active Human Involvement.** For safety-critical tasks such as autonomous driving, the safety of both the controlled vehicles and the human subjects is the top priority. There are many works that allow human subjects to proactively involve the agent-environment interactions based on their own judgment to ensure safety, which we call active human involvement. Human subjects

can terminate the episode if a near-accidental situation happens and such intervention policy can be learned [56, 1, 43, 35, 54, 50]. Recent studies explore active human involvement methods through intervention and demonstration in the human-agent shared autonomy [27, 30, 17, 47, 26, 16, 54]. However, previous methods do not fully utilize the power of human involvement. COACH [27] treats human labels as indications of advantage instead of simply as reward. Compared to COACH, our method accepts not only the feedback (the intervention signal) but also the human demonstration. Our method does not consider the time delay of human subjects explicitly as COACH does. Interactive imitation learning method (HG-DAgger) [17] does not leverage data collected by agents, while Intervention Weighted Regression (IWR) [29] does not suppress undesired actions likely intervened by human. Meanwhile, Expert Intervention Learning (EIL) [47] and IWR [29] focus on optimizing actions step-wise without considering the temporal correlation between steps. These drawbacks harm learning efficiency and thus incur more human involvement. Moreover, previous methods lack experiments to demonstrate the generalizability to different task settings and human control devices.

## 3 Problem Formulation

Policy learning aims at finding a policy to solve the sequential decision-making problem, which is usually modeled by a Markov decision process (MDP). MDP is defined by the tuple $M = \langle \mathcal{S}, \mathcal{A}, \mathcal{P}, r, \gamma, d_0 \rangle$ consisting of a state space $\mathcal{S}$, an action space $\mathcal{A}$, a state transition function $\mathcal{P} : \mathcal{S} \times \mathcal{A} \to \mathcal{S}$, a reward function $r : \mathcal{S} \times \mathcal{A} \to [R_{\min}, R_{\max}]$, a discount factor $\gamma \in (0, 1)$, and an initial state distribution $d_0 : \mathcal{S} \to [0, 1]$. The goal of conventional reinforcement learning is to learn a *novice policy* $\pi_n(a|s) : \mathcal{S} \times \mathcal{A} \to [0, 1]$ that can maximize the expected cumulative return: $\pi_n = \arg\max_{\pi_n} \mathbb{E}_{\tau \sim P_{\pi_n}} [\sum_{t=0}^{T} \gamma^t r(s_t, a_t)]$, wherein $\tau = (s_0, a_0, ..., s_T, a_T)$ is the trajectory sampled from trajectory distribution $P_{\pi_n}$ induced by $\pi_n$, $d_0$ and $\mathcal{P}$. Here $\pi_n$ defines a stochastic policy, while deterministic policy can be denoted as $\mu_n(s) : \mathcal{S} \to \mathcal{A}$ and its action distribution is a Dirac delta distribution $\pi_n(a|s) = \delta(a - \mu_n(s))$.

The reward function imposes an assumption that the reward can fully reflect the intentions of the users and incentivize desired behaviors. However, this assumption may not always hold and the learned agent may obtain biased behaviors or figure out the loophole to finish the task [23, 40]. Revisiting the primal goal when developing learning systems, we find the reward is not a necessity since what we really want to achieve is the realization of human preference in the learned behaviors and, as suggested by [40], the ultimate source of information about human preferences are human behaviors.

Imitation Learning (IL) methods directly learn $\pi_n$ from human behaviors. Assuming a human expert has a *human policy* $\pi_h(a_h|s) : \mathcal{S} \times \mathcal{A} \to [0, 1]$, which outputs human action $a_h \in \mathcal{A}$. Note that human action shares the same action space as novice action. IL learns from the trajectories generated by human policy $\tau_h \sim P_{\pi_h}$ and optimizes the novice policy to close the gap between $\tau_n \sim P_{\pi_n}$ and $\tau_h$. Instead of generating an offline dataset and training novice policy against it [14, 9], we can incorporate a human subject into the loop of training for providing online data. This can mitigate the distributional shift since the data generated with human-in-the-loop has closer state distribution to that of the novice policy [39]. This can be modeled by introducing an *intervention policy* $I(\cdot|s, a_n)$ to describe human subjects' intervention behaviors. In earlier methods such as DAgger [39], the intervention policy is a Bernoulli distribution and the control authority switches back and forth between the novice and the expert. It is unrealistic to invite a real human subject to be involved in such training. Later studies allow the human subjects to intervene and take full control [49, 43, 26, 54], which we call such setting as *learning from active human involvement*. During training, a human subject accompanies the novice policy and can intervene with the agent by taking over the control to demonstrate desired behaviors. The intervention policy can be considered as a deterministic policy denoted by $I(s, a_n) : \mathcal{S} \times \mathcal{A} \to \{0, 1\}$ where $a_n \sim \pi_n(\cdot|s)$ is agent's action. With notations above, the *behavior policy* $\pi_b$ that generates actions during training is:

$$\pi_b(a|s) = (1 - I(s, \mu_n(s)))\delta(a - \mu_n(s)) + I(s, \mu_n(s))\pi_h(a|s). \tag{1}$$

With such a model of active human involvement, we can now formulate our objectives.

**Task-specified metrics.** Our primal goal is to find novice agents whose behaviors are well-aligned with human preferences. In this work, we inform the human subjects of the primal goal of the tasks, *e.g.* navigating to the destination in driving tasks. They are also aware of how task-specified metrics, such as success rate and route completion provided by the test environments, are computed.

These metrics serve as a proxy for human preferences in evaluating trained agents' performance. Unlike prior work where these metrics were used as rewards, our learning agent cannot access them. The only supervision sources in our method are human interventions, $I(s, a)$, and demonstrations, $a_h \sim \pi_h(\cdot|s)$.

**Preference Alignment.** In our method, humans can intervene at any time. Most interventions occur in near-accidental situations or when agents are performing poorly. Conversely, lack of intervention indicates alignment with human preferences. Hence, another goal is to develop a novice policy that minimizes human interventions during shared control. In the next section, we will discuss our insights and how we build a concise, general, and efficient learning method to achieve these objectives.

## 4    Method

We propose the *Proxy Value Propagation (PVP)* method which can transform a value-based RL method into an efficient reward-free human-in-the-loop policy optimization method that learns from active human involvement. PVP is compatible with various task settings, such as continuous and discrete action spaces, as well as various human control devices. In this section, we first summarize the basic workflow of value-based RL before introducing the motivation and the design of PVP. We then describe the implementation details.

**Value-based RL:**  The proposed human-in-the-loop method results from the minimum modification of existing reinforcement learning methods. Thus, we briefly introduce the background of related methods. Value-based RL optimizes the value function and policy iteratively. On the value function side, we denote the state-action value and state value of policy $\pi$ as $Q(s, a) = \mathbb{E}\left[\sum_{t=0}^{\infty} \gamma^t r(s_t, a_t)\right]$ and $V(s) = \mathbb{E}_{a \sim \pi(\cdot|s)} Q(s, a)$, respectively. A neural network is commonly used to estimate the value function with Bellman backup: $Q(s, a) \leftarrow r(s, a) + \gamma \max_{a'} Q(s', a')$, where $s'$ is the next state. To learn the value network $Q_\theta$ parameterized by $\theta$, stochastic gradient descent on the temporal difference (TD) loss is conducted $J^{\text{TD}}(\theta) = \mathbb{E}_{(s,a,s')} |Q_\theta(s, a) - (r(s, a) + \gamma \max_{a'} Q_{\hat{\theta}}(s', a'))|^2$, where $Q_{\hat{\theta}}$ can be a delay-updated target network. In this work, we adopt the TD learning in the **reward-free** setting. Remove the reward in the TD loss, the TD loss becomes:

$$J^{\text{TD}}(\theta) = \mathbb{E}_{(s,a,s')} |Q_\theta(s, a) - \gamma \max_{a'} Q_{\hat{\theta}}(s', a')|^2. \qquad (2)$$

On the policy side, based on the learned value function, the deterministic policy $\mu_n$ parameterized by $\phi$ can be learned by maximizing the Q values: $J(\phi) = \mathbb{E}_s Q(s, \mu_n(s; \phi))$. The optimal policy is expected to maximize Q values:

$$\mu_n(s) = \arg\max_a Q(s, a). \qquad (3)$$

### 4.1    Proxy Value Propagation

We illustrate the active human involvement of PVP in Fig. 1. During training, the human subject supervises the agent-environment interactions (Fig. 1 **A**). Those exploratory transitions by the agent are stored in the Novice Buffer $\mathcal{B}_n = \{(s, a_n, s')\}$. At any time, the human subject can intervene the free exploration of the agent by pressing a button in the control device (Fig. 1 **B**). While pressing the button, the human takes over the control and provides a demonstration of how to behave. During human involvement, both human and novice actions will be recorded into the Human Buffer $\mathcal{B}_h = \{(s, a_n, a_h, s')\}$. Concurrently with the human-agent shared control, our method keeps updating the novice policy by the novel Proxy Value Propagation mechanism (Fig. 1 **C**), which will be discussed later.

In the shared human-agent control, human intervention serves as a distinct indicator of suboptimal agent performance, which could result from the agent executing perilous actions or exhibiting ineffective behaviors. Thus, the optimal policy learned by the agent should (1) strive to approximate the behaviors demonstrated by the human subjects and (2) avoid performing actions that are intervened by humans.

The key insight of this work is that we can manipulate the Q values to induce desired behaviors, given that value-based RL has the nature to seek value-maximizing policy as Eq. 3. As shown in Fig. 1 **C**, for emulating human behavior and minimizing intervention, we sample data $(s, a_n, a_h)$ from the

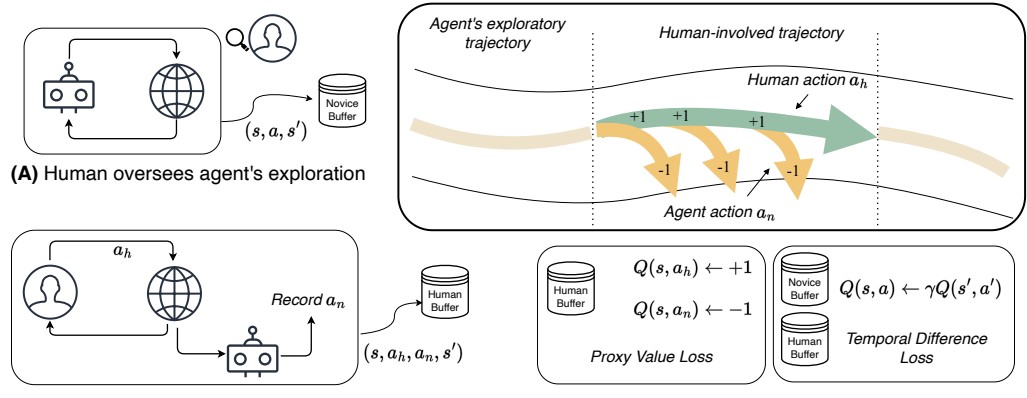

Figure 1: Illustration of Proxy Value Propagation. **(A)** Human oversees the agent's trial-and-error exploration with the environment. When the human subject does not intervene, the transitions will be recorded into the novice buffer $\mathcal{B}_n$. **(B)** When the human intervenes, both novice action $a_n$ and human action $a_h$ will be recorded into the human buffer $\mathcal{B}_h$ but only the human action will be applied to the environment. **(C)** In training, we use the human buffer to compute proxy value loss and propagate the human intent knowledge to all transitions via TD loss without access to the reward.

human buffer and label the Q value of the human action $a_h$ with $+1$ and the novice action $a_n$ with $-1$. This is achieved by fitting the Q network directly with PV loss:

$$J^{\text{PV}}(\theta) = \mathop{\mathbb{E}}_{(s,a_n,a_h)}[|Q_\theta(s,a_h) - 1|^2 + |Q_\theta(s,a_n) + 1|^2]I(s,a_n). \tag{4}$$

The transitions in the novice buffer are not intervened by the human subject, meaning they are aligned with human preferences. Meanwhile, those transitions also contain information of the forward dynamics [24, 55]. To exploit the information contained in these transitions, instead of discarding these data as in [17], we propagate the proxy values to these states via TD learning in Eq. 2 and use those transitions together with those human-involved transitions for the policy learning. The final value loss is evaluated as follows:

$$\begin{aligned} J(\theta) = J^{\text{PV}}(\theta) + J^{\text{TD}}(\theta) = \mathop{\mathbb{E}}_{(s,a_n,a_h)\sim\mathcal{B}_h}&[|Q_\theta(s,a_h) - 1|^2 + |Q_\theta(s,a_n) + 1|^2]I(s,a_n) \\ &+ \mathop{\mathbb{E}}_{(s,a,s')\sim\mathcal{B}_h\bigcup\mathcal{B}_n}|Q_\theta(s,a) - \gamma\max_{a'}Q_{\hat{\theta}}(s',a')|^2 \end{aligned} \tag{5}$$

Then we follow the policy update process outlined in the base RL methods.

## 4.2 Analysis

**Connection to CQL.** The proposed PVP method can be interpreted as adopting the Conservative Q-Learning (CQL) [21] objective for reward-free and online learning settings. It augments the CQL objective with an extra L2 regularization term imposed on the Q-values for human-involved transitions. In our online learning setting, Eq. 5 can be reformulated as:

$$J(\theta) = \mathop{\mathbb{E}}_{\mathcal{B}_h,I(s,a_n)=1}\underbrace{[Q_\theta^2(s,a_n) + Q_\theta^2(s,a_h)}_{\text{L2 Regularization on Q}} + 2 + \underbrace{2(Q_\theta(s,a_n) - Q_\theta(s,a_h))]}_{\text{CQL Loss}} + \text{TD loss.} \tag{6}$$

CQL was originally proposed to mitigate the problem of overestimated Q-values in offline RL settings. These overestimations often lead to suboptimal policies due to the optimistic selection of actions with misleadingly high values. In our work, we deal with human actions and novice actions sampled from two different distributions, where overestimation might also occur. However, unlike CQL, PVP does not have access to a reward function, meaning the Q-values are not grounded in an estimation of true values. The additional L2 regularizer therefore serves to impose constraints on the Q-values, helping to prevent unbounded growth and potential overfitting. In Sec. 5.4, we compare the learned proxy Q-values under both CQL and PVP objectives. Our results indicate that human and agent actions are more distinguishable when learned through PVP.

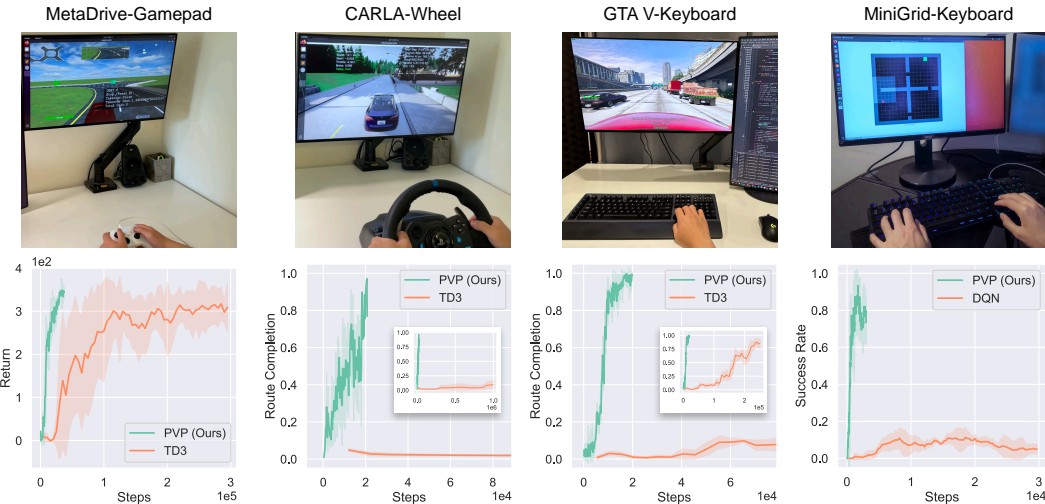

Figure 2: Evaluation of PVP under four different environments with human control devices. For each environment, we plot the test-time performance curve of the agent trained by the proposed PVP and the RL counterpart TD3. The x-coordinate is the total number of environment interactions, which indicates the time steps the training agent (in RL method) or the human-agent system (in our method) experiences during training. Compared to the RL counterpart, the proposed method achieves much higher performance with superior learning efficiency.

**Alternative to Reward Assignment.** On the other hand, a more straightforward idea than PVP is to assign a reward of +1 to human actions and -1 to agent actions during intervention. Unfortunately, it is not practical since the Bellman backup is conducted on the transition triplet $(s, a, s')$, where one has to use future states' values to estimate current values. Therefore, the reward must correspond to the action from the behavior policy, the action $a$ causes the transitions from $s$ to $s'$. In our context, during involvement, the action $a = a_h$ must come from human policy as the human subject is taking control. Though we can assign $+1$ reward and compute value target in those human-involving transitions, we have no way to assign $-1$ to the agent's actions because we don't know the next states caused by those actions and thus we can't compute the value target. It is also not practical to query the environment to get the next state $s'' \sim \mathcal{P}(s, a_n)$ as the $a_n$ is a potentially danger or undesired action and replaying it in the real-world environment is not feasible. In our preliminary experiment, we find the policy fails to learn anything regardless of the amount of human involvement provided. This is because the reward will be $+1$ for all the human-involving transitions and the learning agent will find a pitfall to maximize its rewards: it always demonstrates undesired behaviors so that humans will always take control, which yields a $+1$ reward.

### 4.3  Implementation Details

**Base RL Methods.** Our method can be implemented for both continuous and discrete action spaces by extending TD3 [10] and DQN [31] with PV loss and the balanced buffer. While TD3 uses a deterministic policy, DQN adopts epsilon-greedy exploration that makes the policy stochastic. We remove the action noise in DQN and simply follow the argmax rule to select actions. Therefore, our method enjoys deterministic novice policy in both cases. The primary reason is that according to the feedback of human subjects, stochastic novice makes human subjects experience excessive fatigue due to the difficulty in monitoring and correcting agents' noisy actions. This design choice makes our method more user-friendly, as shown in the user study in Sec. 5.3.

**Balanced Buffers.** The intervention gradually becomes sparse as the agent learns to reduce human intervention. However, those sparse intervention signals contain even more important information on how to behave under critical situations. Previous method [26] stores agent data and human data into one buffer and samples them uniformly. Abusing the notations, the ratio between the transitions from the agent's exploration and from human involvement is $|\mathcal{B}_n| : |\mathcal{B}_h|$ in each SGD batch. The human demonstrations are overwhelmed by the amount of agent-generated trajectories,

Table 1: Comparison of different approaches in MetaDrive-Keyboard. The overall intervention rate is given besides the human data usage.

| Method | Training | | | Testing | | |
|---|---|---|---|---|---|---|
| | Human Data Usage | Total Data Usage | Total Safety Cost | Episodic Return | Episodic Safety Cost | Success Rate |
| SAC | - | 1M | 2.76K $\pm$ 0.95K | 386.77 $\pm$35.1 | 0.73 $\pm$1.18 | 0.82 $\pm$0.18 |
| PPO | - | 1M | 24.34K $\pm$3.56K | 335.39 $\pm$12.41 | 3.41 $\pm$1.11 | 0.69$\pm$0.08 |
| TD3 | - | 1M | 1.74K $\pm$ 0.62K | 318.12 $\pm$21.9 | 0.47 $\pm$0.08 | 0.70 $\pm$0.09 |
| SAC-Lag | - | 1M | 1.84K $\pm$ 0.49K | 351.96 $\pm$101.88 | 0.72 $\pm$0.49 | 0.73 $\pm$0.29 |
| PPO-Lag | - | 1M | 11.64K $\pm$ 4.16K | 299.99 $\pm$49.46 | 1.18 $\pm$0.83 | 0.51 $\pm$0.17 |
| CPO | - | 1M | 4.36K $\pm$2.22K | 194.06 $\pm$108.86 | 1.71 $\pm$1.02 | 0.21 $\pm$0.29 |
| Human Demo. | 30K | - | 39 | 347.523 | 0.39 | 0.97 |
| BC | 30K (1.0) | - | - | 113.32 $\pm$10.21 | 2.171 $\pm$0.65 | 0.073 $\pm$0.02 |
| GAIL | 30K (0.015) | 2M | 25.90K $\pm$ 8.15K | 81.51 $\pm$ 9.43 | 1.308 $\pm$ 0.23 | 0.0 $\pm$ 0.0 |
| HG-Dagger | 39.0K (0.76) | 51K | 56 | 116.393 | 1.979 | 0.045 |
| IWR | 35.8K (0.79) | 45K | 52 | 226.221 | 1.457 | 0.465 |
| HACO | 19.2K (0.48) | 40K | 130 | 143.287 | 1.645 | 0.139 |
| PVP w/o TD | 13.5K (0.34) | 40.5K | 70 | 252.447 | 1.277 | 0.220 |
| PVP w/ Reward | 12.8K (0.32) | 40K | 30 | 319.383 | 0.767 | 0.755 |
| PVP (Ours) | 14.6K (0.37) | 40K | 76.8 $\pm$9.3 | 353.636 $\pm$23.7 | 0.898 $\pm$0.15 | 0.857 $\pm$0.04 |

leading to inefficient learning of those critical human behaviors and even catastrophic forgetting. For example, the driving policy sometimes fails to master acceleration at the beginning of an episode, even though the human subject has already demonstrated the expected maneuver multiple times. This is because the demonstration of initial acceleration only lasts a short period of time and thus is scarce in the buffer. To address this issue, we balance the transitions coming from the human buffer and the novice buffer. In each training iteration, we sample two equally-sized batches $b_n$ and $b_h$ from $\mathcal{B}_n$ and $\mathcal{B}_h$ respectively, each has $N/2$ samples. $N$ is the batch size for the policy update. By concatenating $b_n$ and $b_h$, our method can balance the data from the human's demonstration and from the agent's exploration, and hence the ratio between two types of data in each SGD batch keeps $1:1$. Therefore, in the initial acceleration example above, the balanced buffer recalls the acceleration behavior, preventing catastrophic forgetting.

## 5 Experiments

### 5.1 Experimental Setting

**Tasks.** We conduct experiments on various control tasks with different observation and action spaces. For continuous action space, we use three driving environments, MetaDrive safety benchmark [25], CARLA Town01 [8], and a customized driving environment built upon Grand Theft Auto V (GTA V), a popular video game. In these tasks, the agent needs to steer the target vehicle with low-level acceleration, braking, and steering, to reach its destination. Specifically, in MetaDrive safety environments, the agent needs to avoid any crash in the heavy-traffic scene with normal vehicles, obstacles, and parked vehicles. In MetaDrive, there exists a split of training and test environments, and we present the performance of the learned agent in a held-out test environment. To examine our method with different observation modalities, we use the sensory state vector in MetaDrive and GTA V and the bird-eye view image in CARLA as observation. For discrete action space, we use MiniGrid Two Room task [4], which involves agent exploration such as moving toward a door and opening the door before reaching the destination. The observation of MiniGrid is the semantic map of the agent's local neighborhood. Please refer to Appendix E for more information about the environment setup.

**Evaluation Metrics.** In MetaDrive safety benchmark, we report *total safety cost* as the number of crashes during training, which reflects the number of potential dangers exposed to the human subject during training. We also report *episodic return*, *episodic safety cost*, and *success rate* as the test performance of the agents. Episodic safety cost is the average number of crashes in one

Table 2: Results of different approaches in CARLA.

| Method | Human Data | Total Data | Route Completion | Success Rate |
|--------|-----------|-----------|------------------|--------------|
| PPO | - | 1M | $0.24 \pm 0.013$ | $0.0 \pm 0.0$ |
| TD3 | - | 1M | $0.11 \pm 0.05$ | $0.0 \pm 0.0$ |
| BC | 5K | - | $0.42 \pm 0.08$ | $0.20 \pm 0.10$ |
| HG-DAgger | 6.8K | 24K | 0.64 | 0.47 |
| IWR | 5.7K | 24K | 0.69 | 0.60 |
| HACO | 4.8K | 24K | 0.52 | 0.40 |
| PVP (Ours) | 6.6K | 24K | $0.92 \pm 0.05$ | $0.73 \pm 0.08$ |

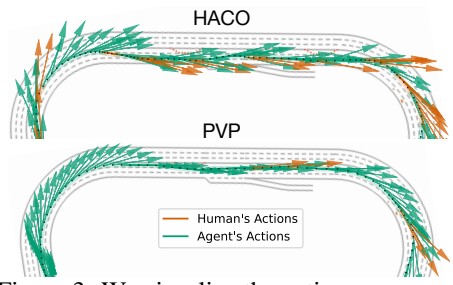

Figure 3: We visualize the action sequences generated by HACO and PVP agents in the same MetaDrive map who are trained to 40K steps. PVP has much smoother actions.

episode. The success rate is the ratio of episodes in which agents reach the destination to the total test episodes. In CARLA, we report *route completion* and *success rate*. Route completion is the ratio of the traveled distance to the length of the complete route. GTA V uses *route completion* and MiniGrid uses *success rate* to measure the performance. Except for total safety cost, the aforementioned metrics measure the test-time performance, which is tested when the agent runs independently without human involvement. For human-in-the-loop experiments, we also report the total number of human-involved transitions (*human data usage*) and the *overall intervention rate*, which is the ratio of human data usage to total data usage. These show how much effort humans make to teach the agents. We also design a user study to measure the experience of human subjects in Sec. 5.3.

**Human Interfaces.** To examine the generalizability of our method, we leverage multiple control devices: Xbox Wireless Controller (Gamepad), keyboard, and Logitech G29 Racing Wheel. We denote the MetaDrive tasks with three devices as MetaDrive-Gamepad/Keyboard/Wheel. As shown in Fig. 2, human subjects can takeover through control devices and monitor the training process through the visualization of environments on the screen. The Ethics statement is provided in Appendix A.

**Experimental Details.** We implement most of the code with Stable-Baselines3 [37]. Training results of various baselines in MetaDrive tasks are obtained from the open-source code by [26]. The RL baselines are repeated 5 times with different random seeds, while other human-in-the-loop methods are repeated fewer times due to limited human resources. In the training of the human-in-the-loop methods, a real human subject participates in each experiment and we do not use any simulated user input. During testing, there is no form of human involvement. For each experiment, we evaluate each checkpoint in the environment for multiple runs and report the average task-specified metrics as the performance of this checkpoint. We report the performance of the best checkpoint as the result of the experiment. We provide the standard deviation if the experiments are repeated multiple runs in tables and figures. All experiments with humans are conducted on a local computer with an Nvidia GeForce RTX 3080. The local computer can support real-time simulation and training. Hyper-parameters and other details are given in Appendix E and G.

**Baselines.** We test four native RL baselines: PPO [45], SAC [13], TD3 [10] and DQN [31]. We also test three safe RL baselines: Constraint Policy Optimization (CPO) [2], PPO-Lagrangian [48], SAC-Lagrangian [12]. In all baselines above, the reward function and cost function (for MetaDrive Safety Benchmark) are defined by the environment and can be accessed by the agents. We also test IL methods Behavior Cloning (BC) and GAIL [14]. Human-in-the-loop methods that learn from active human involvement are tested: Human-Gated DAgger (HG-DAgger) [17], Intervention Weighted Regression (IWR) [29] and Human-AI Copilot Optimization (HACO) [26].

## 5.2 Baseline Comparison

**Comparing with RL Counterparts.** Fig. 2 shows the curves of test-time performance. In MetaDrive-Gamepad, our method achieves 350 returns in 37K steps. This takes about one hour in the real-world HL experiment. TD3 baseline fails to achieve comparable results even after 300K steps of training. In CARLA, PVP agents learn to drive within 30 minutes with our method, while TD3 cannot solve the task. In GTA V, PVP can solve the task with 1.2K human data usage and 20K total data usage. The whole experiment takes only 16 minutes. TD3 instead utilizes 300K steps to achieve similar

Table 3: User study result. The maximum score for each item is 5.

|            | HG-DAgger      | IWR            | HACO           | PVP            |
|------------|----------------|----------------|----------------|----------------|
| Compliance | $3.0 \pm 0.8$  | $4.0 \pm 0.8$  | $3.0 \pm 0.2$  | $4.8 \pm 0.5$  |
| Performance| $2.2 \pm 1.0$  | $3.7 \pm 0.9$  | $3.3 \pm 0.9$  | $4.8 \pm 0.5$  |
| Stress     | $3.2 \pm 0.9$  | $4.5 \pm 0.5$  | $2.3 \pm 0.9$  | $4.7 \pm 0.6$  |

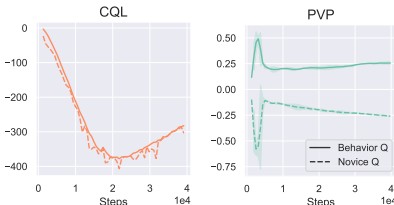

Figure 4: Evolution of proxy values.

performance. In MiniGrid tasks, our method successfully solves the tasks while vanilla DQN fails, showing that PVP can learn an exploratory solution and can be incorporated into discrete action space. We also show experiments on one easier and one harder MiniGrid environment in Appendix F.2, where PVP greatly improves learning efficiency.

**Comparing with Human-in-the-loop Baselines.** Table 1 suggests all tested HL methods achieve extremely low safety violations in training compared to vanilla RL and Safe RL methods, empirically supporting the preference alignment of the active human involvement, if we consider human preference is to avoid safety violation. Compared to other human-in-the-loop methods, our method costs the lowest human efforts in terms of human data usage and overall intervention rate, while greatly outperforming baselines in testing performance. Since MetaDrive has a training and test environment split, the result suggests PVP can learn high-quality agents with generalizability. Similar results are shown in CARLA in Table 2. Compared to RL baselines, HL methods achieve decent success rates and route completion rates even with only 24K environmental interactions. Compared to HL baselines, PVP achieves the best route completion rate.

**Visualization.** In Fig. 3, we visualize the action sequences of the agents trained by PVP and a human-agent shared control baseline HACO [26]. The angle and length of each arrow represent the steering and acceleration, respectively. The human subject's actions are marked with yellow. Compared to HACO method, PVP agent produces smoother actions, which explains its high user study scores shown in the next section.

### 5.3   User Study

We design a user study questionnaire to assess the experience of human subjects. Details are provided in the human subject research protocol in Appendix B. Three aspects are considered: (1) **Compliance** measures whether the behaviors of the agent satisfy human intents. For example, a highly compliant agent behaves like human such that the human subjects feel like they are completing objectives by themselves. (2) **Performance** is the subjective evaluation from human subjects on whether the agent can solve the primal task, e.g. driving to the destination in navigation tasks. This score should be low if the agent cannot learn a particular behavior or forgets it even though human subjects have taught the agent multiple times. (3) **Stress** gauges the cognitive cost human subjects pay to train the agent. A typical source of stress is the annoying oscillation and jitter the agent demonstrates. Unexpected behavior that requires human's instant reaction also creates stress. A lower score means more stress.

Table 3 shows our method is the most user-friendly method. On the one hand, we use a deterministic novice policy that greatly alleviates the jitter and unexpected behaviors, reducing stress. On the other hand, our method masters human behaviors and suppresses undesired actions with the balanced buffer and proxy value, improving the user experience in compliance and performance.

### 5.4   Ablation Studies

**TD learning:** As shown in Table 1 "PVP w/o TD", disabling TD learning via setting $J^{\text{TD}}(Q) = 0$ significantly damages the performance of PVP, suggesting that propagating information from human-involved states to other states is critical to the success of PVP.

**PVP with reward:** Both MetaDrive and CARLA results in Table 1 and 4 show that adding the environmental reward doesn't bring significant improvement in the learning performance, which might be caused by the fact that the native reward function might not be aligned with human preference.

**Balanced buffer:** We find that disabling balanced buffers (PVP w/o BB) makes the training unstable and leads to poor performance. This design avoids the catastrophic forgetting when the agent-generated data overwhelms the human demonstrations as in HACO [26].

**Novice buffer:** We find that PVP without the Novice buffer (PVP w/o NB) yields poor performance. The agent data stored in the novice buffer contains information on human preference and the forward dynamics of the environment. Thus, PVP does not discard the agent exploratory data as opposed to HG-DAgger [17].

Table 4: Ablation studies in CARLA.

| Method | Human Data | Route Completion | Success Rate |
|---|---|---|---|
| HACO | 4.8K | 0.52 | 0.40 |
| HACO w/o SP | 5.1K | 0.49 | 0.20 |
| PVP w/o BB | 2.8K | 0.62 | 0.33 |
| PVP w/o NB | 4.2K | 0.708 | 0.33 |
| PVP w/ Rew. | 4.4K | 0.793 | 0.467 |
| PVP w/ SP | 12.3K | 0.40 | 0.20 |
| PVP w/ CQL | 8.0K | 0.622 | 0.266 |
| PVP (Ours) | 6.6K | $0.92_{\pm 0.05}$ | $0.73_{\pm 0.08}$ |

**Stochastic policy:** We implement PVP based on Soft Actor-critic [13] so that the novice policy is now a stochastic policy. As shown in the "PVP w/ SP" in Table 4, introducing randomness in novice actions greatly reduces the performance. The human subjects report that the novice agents with stochastic policy oscillate frequently, making it hard to respond when the agents suddenly drive toward the side road. HACO [26] has similar human-AI shared control as PVP, but it adopts a stochastic policy. For comparison, we also implement HACO without a stochastic policy. "HACO w/o SP" suggests deterministic policy can not bring significant improvement to HACO.

**Regularization on Q values**: As discussed in Sec. 4.2, PVP objective can be interpreted as CQL with a newly introduced L2 regularization term on the Q values. We conduct the experiment to evaluate the performance of the vanilla CQL objective with other PVP designs in our reward-free online learning settings. As shown in "PVP w/ CQL" in Table 4, CQL objective yields worse performance. This experiment shows that the vanilla CQL doesn't work in this human active involvement setting. As shown in Fig. 4, the proxy value in the vanilla CQL method has a much larger magnitude which makes the values of behavior actions (the actions applied to the environment) and agent actions hard to distinguish. PVP has smoother proxy values with a clear margin between behavior and novice Q. CQL does not set a bound for the proxy value, thus proxy values in those extreme human actions are reinforced without a bound, making the novice policy rapidly learn those extreme actions, whereas PVP has bounded proxy values, leading to more stable training and better overall performance.

# 6 Conclusion

Learning through active human involvement is a promising approach enabling safe and efficient policy learning. In this work, we propose *Proxy Value Propagation (PVP)* that can effectively learn from the intervention and the corrective feedback from active human involvement. PVP can be seamlessly integrated into existing value-based RL methods and achieves highly efficient reward-free policy learning, without offline pretraining and reward engineering. Human-in-the-loop experiments show the proposed method achieves superior performance and better user experience across diverse environments with different action spaces and human control devices, showing that the learning from active human involvement is a efficient policy learning method aligning human preference.

**Limitations.** (1) We only apply our method to two value-based RL methods. Advanced techniques such as exploration encouraging [33] and prioritized replay buffer [44] can be added to further improve the result. (2) Our method is not applicable to tasks where humans can not provide demonstrations. (3) We assume that human always demonstrates desired actions. We will show in Appendix D that suboptimal human behaviors will damage learning. In this case, we can define a sparse cost function in the training environment and utilize constrained optimization [2] to penalize bad demonstrations. (4) We assume that human subjects are available and attentive throughout the entire training. While our method is proven to be effective even under heavy traffic environments, we plan to further enhance its sample efficiency. We will achieve this goal by conducting offline RL training and policy evaluation in the background or passively involving human subjects whenever the model is uncertain about the environment.

**Acknowledgment**: This work was supported by the National Science Foundation under Grant No. 2235012. The human experiment in this study is approved through the IRB#23-000116 at UCLA.

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

# A  Ethics Statement

Human subjects get paid to participate in the experiments. They can pause or stop the experiment if any discomfort happens. No human subjects are injured because all tasks we test are in virtual simulation. Each experiment will not last longer than one hour and subjects will rest at least three hours after one experiment. During training and data processing, no personal information is revealed in the collected dataset or the trained agents. We have obtained IRB approval to conduct this project.

# B  Human Subject Research Protocol

**Recruiting and Requirement.** For our study, we recruit 5 human subjects. All of them are college students and have the age from 20 to 30 years. Furthermore, every participant are required to have a valid driver's license and have experience in playing video game. Participation in our study is entirely voluntary. We ensure transparency by informing all subjects about the nature of the experiments and how their demonstrations would be used. Every subject provide written consent, confirming they are fully aware and in agreement. Additionally, the study is conducted with the IRB approval.

**Onboarding Period.** Participants are required to undergo a practice session, during which they drive under complete control to get a sense of the control devices (wheel, gamepad and keyboard), the environment interface, the dynamics of each environment and how an episode will fail or success. Each subject get familiar with all the control devices and all the environments, which is indicating by performing at least 10 successful episodes, before they participate in the main experiments.

**Main Experiment.** During the initial stages of the formal experiment, subjects are *advised* to retain full control of the agent for the first one or two episodes. Subsequently, they may begin to let the agents taking control and intervene as necessary. The objective in all driving experiments is twofold: firstly, to safely navigate the vehicle to its designated destination, and secondly, to ensure the vehicle's operation aligns with traffic regulations and human preferences.

Subjects are encouraged to perform intervention whenever they perceive the vehicle might be in a dangerous situation, in violation of traffic rules, or in whatever scenario the human subjects feel they wouldn't behave in the way the novice policies do.

To ensure data integrity and counter potential proficiency biases, the order of experiments with different control devices, tasks and training algorithms is randomized for each subject. By doing this we mitigate the bias that a subject might become more familiar with the task when experimenting different algorithms.

**User Study Questionnaire.** We design a user study questionnaire to assess the experience of human subjects. The questionnaire is provided in Appendix B. Three aspects are considered:

- **Compliance** measures whether the behaviors of the agent satisfy human intents. For example, a highly compliant agent behaves like human such that the human subjects feel like they are completing objectives by themselves.

- **Performance** is the subjective evaluation from human subjects on whether the agent can solve the primal task, e.g. driving to the destination in navigation tasks. This score should be low if the agent cannot learn a particular behavior or forget it even though human subjects have taught the agent multiple times.

- **Stress** gauges the cognitive cost human subjects pay to train the agent. A typical source of stress is the annoying oscillation and jitter the agent demonstrates. Unexpected behavior that requires human's instant reaction also creates stress. A lower score means more stress.

The same questions are repeated for each algorithm the human subject experimenting on.

**Compliance**: Generally, do you think the agent trained with this method complies with your intention? The higher score the better.
Examples:
(+) Good: Agent drives as you so that you don't even need to take over.
(-) Bad: Sudden unexpected behavior makes you mad.
Choices: 1, 2, 3, 4, 5

**Performance**: Do you think the agent trained with this method learns fast and performs well in terms of solving the task? The higher score the better.
Examples:
(+) Good: The agent learns fast so I don't need to take over too much in the later period.
(-) Bad: The agent forgets what it learns so I have to re-teach it.
(-) Bad: The agent never learns a specific behavior like accelerating or turning even though I have taught it so many times.
Choices: 1, 2, 3, 4, 5

**Stress**: Do you think training with this agent is tired or stressed? The lower score the more fatigue and stress. A higher score means you are more relaxed.
Tiredness might come from many sources: Oscillating trajectory, unexpected behaviors, degrading performance that you have to re-teach, etc.
It is possible that your agent is not performing well but you don't feel tired training it. On the other hand, it is possible that your agent has good performance but still causes fatigue due to unexpected behaviors.
Choices: 1, 2, 3, 4, 5

## C  Demo Video

Please find our demo video in the supplementary material. This video shows the footage of human experiments and the comparisons between agents learned by the baselines and the proposed method. The video contains three sections:

1. The first section shows how we learn the driving policy in CARLA task within 20 minutes. We also compare the behavior of agents learned from PVP and TD3 baseline.

2. In the second section, we show the footage of MetaDrive human experiment where the human subject uses a gamepad as the control device. We present the behavior comparison between PVP and TD3 baseline.

3. In the third section, we show the applicability of our method to other tasks. PVP performs well in GTA V and can drive smoothly on the highway. In the discrete control tasks, the behavior comparison between PVP and DQN baseline in MiniGrid Empty Room and Four Room are provided.

## D  Preference Alignment

Here we provide a conceptual framework to describe the compliance of human intention. First, we introduce a ground-truth indicator $C : \mathcal{S} \times \mathcal{A} \to \{0, 1\}$ of the intention violation, denoting whether the action is undesired. $C$ is not revealed to the learning algorithm.

$$C(s, a) = \begin{cases} 1, & \text{if } a \text{ violates human intention} \\ 0, & \text{otherwise.} \end{cases} \tag{7}$$

We will derive the upper bound of the discounted occurrence of intent violation, a measure of training time human intent compliance:

$$S_{\pi_b} = S_{\pi_b}(s_0) = \mathbb{E}_{\tau \sim P_{\pi_b}} \sum_t \gamma^t C(s_t, a_t), \tag{8}$$

where $P_{\pi_b}$ denotes the probability distribution of trajectories deduced by the behavior policy $\pi_b$.

During training, a human subject shares control with the learning agent. The agent's policy is a deterministic policy $\mu_n(s)$, the human's policy is a stochastic policy $\pi_h(a|s)$. The human subject intervenes $I(s, a) = \text{True}$ under certain state and agent's action $a_n$. The mixed behavior policy $\pi_b$ that produces the real actions to the environment is denoted as:

$$\pi_b(a|s) = (1 - I(s, \mu_n(s)))\delta(a - \mu_n(s)) + I(s, \mu_n(s))\pi_h(a|s), \tag{9}$$

where we use Dirac delta distribution to represent the deterministic novice policy.

Two important assumptions on the human subject are introduced:

**Assumption D.1** (Error rate of human policy). During human-AI shared control, the probability that the human subject produces an undesired action is bounded by a small value $\epsilon < 1$:

$$\mathbb{E}_{s \sim P_{\pi_b}, a \sim \pi_h(\cdot|s)} C(s, a) \le \epsilon. \tag{10}$$

**Assumption D.2** (Error rate of intervention policy). During human-AI shared control, the probability that the human subject does not intervene when the action is undesired is bounded by a small value $\kappa < 1$:

$$\mathbb{E}_{s \sim P_{\pi_b}} (1 - I(s, \mu(s)))C(s, \mu(s)) \le \kappa. \tag{11}$$

We introduce the following theorem and give the proof as follows.

**Theorem D.3** (Upper bound of intent violation). *The discounted occurrence of intent violation $S_{\pi_b}$ of the behavior policy $\pi_b$ is bounded by the error rate of the human action $\epsilon$, the error rate of the human intervention $\kappa$ and the intervention rate $\psi = \mathbb{E}_{s \sim P_{\pi_b}} I(s, a_n)$:*

$$S_{\pi_b} \le \frac{1}{1 - \gamma}(\kappa + \epsilon\psi). \tag{12}$$

*Proof.* Consider Eq. 9, we have:

$$\mathbb{E}_{s \sim P_{\pi_b}, a \sim \pi_b(\cdot|s)} C(s, a) = \mathbb{E}_{s \sim P_{\pi_b}} \{[1 - I(s, \mu_n(s))]C(s, \mu_n(s)) + I(s, \mu_n(s)) \mathbb{E}_{a \sim \pi_h(\cdot|s)} C(s, a)\}$$
$$\le \kappa + \epsilon \mathbb{E}_{s \sim P_{\pi_b}} I(s, \mu_n(s)) = \kappa + \epsilon\psi \tag{13}$$

The upper bound of $S_{\pi_b}$:

$$S_{\pi_b} = \mathbb{E}_{\tau \sim P_{\pi_b}} \sum_{t=0} \gamma^t C(s_t, a_t) \le \sum_{t=0} \gamma^t(\kappa + \epsilon\psi) = \frac{1}{1 - \gamma}(\kappa + \epsilon\psi) \tag{14}$$

$\square$

# E   Environment Details

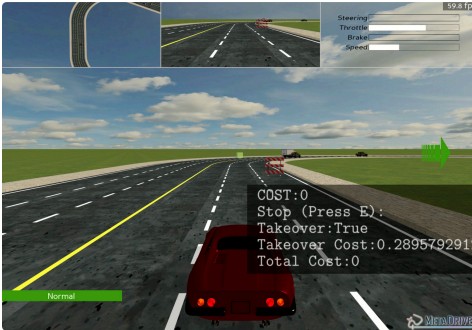

Figure 5: MetaDrive Safety benchmark.

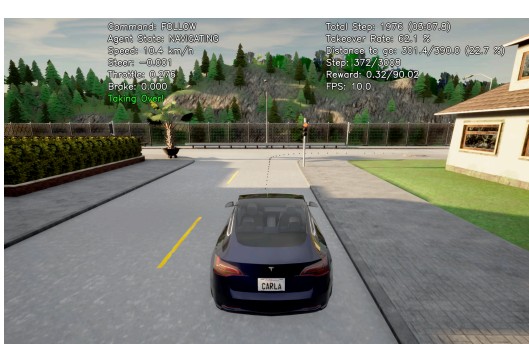

Figure 6: CARLA Town01.

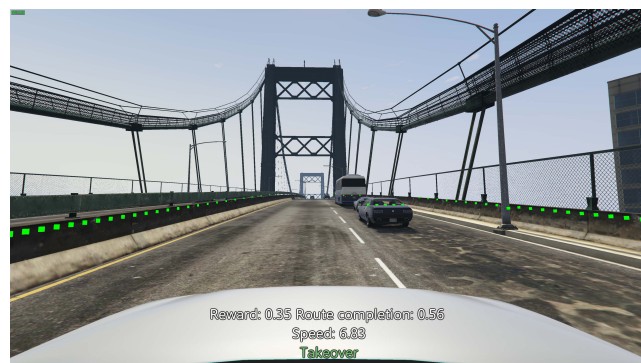

Figure 7: GTA V Training Environment.

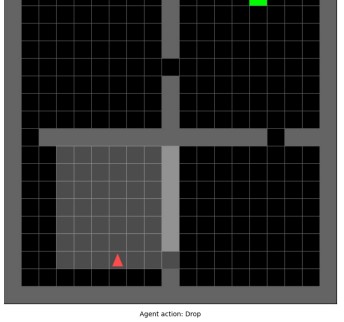

Figure 8: MiniGrid (Four Room).

Table 5: Summary of the experiment environments.

| Environment | Human Input Device | Observation Format | Action Space | Training & Test Set Split |
|---|---|---|---|---|
| MetaDrive | Gamepad, Keyboard, Wheel | State Vector | Continuous | Yes |
| CARLA | Wheel | Bird-eye View Image | Continuous | No |
| GTA V | Keyboard | State Vector | Continuous | Yes |
| MiniGrid | Keyboard | Semantic Map | Discrete | No |

To avoid the potential risks of employing human subjects in physical experiments, we benchmark different approaches in four virtual simulated environments. We conduct experiments on various tasks with different observation and action spaces and human input devices. Table 5 summarizes the differences.

**Continuous action space environments.** For continuous action space, we use MetaDrive Safety Benchmark [25], CARLA Town01 environment [8] and a customized policy learning environment built upon Grand Theft Auto V (GTA V). In these tasks, the agent needs to steer the target car with low-level acceleration, brake and steering and move toward the destination.

MetaDrive Safety Benchmark preserves the capacity to evaluate the safety and generalizability in unseen environments since it uses procedural generation to synthesize an unlimited number of driving maps for the split of training and test sets, which is useful to benchmark the generalization capability of different approaches in the context of safe driving. We train agents in the training set, which contains 50 different scenes, and roll out the learning agents in the test set, which contains another 50 unique scenes. At each episode, the scene (road network) and the spawn location of traffic vehicles and ego vehicle are randomized. We use sensory state vector in MetaDrive as the observation for agents and thus apply MLP network architecture. When running pure RL methods in MetaDrive Safety Benchmark, a -1 penalty will be added to the reward when a crash happens. This is for a fair comparison with the safe RL methods who have access to the cost function directly. Specifically, we follow the default reward scheme in MetaDrive. The reward function in MetaDrive safety benchmark is composed of four parts as follows:

$$R = c_{disp}R_{disp} + c_{speed}R_{speed} + c_{collision}R_{collision} + R_{term}. \qquad (15)$$

1. The displacement reward: $R_{disp} = d_t - d_{t-1}$, wherein the $d_t$ and $d_{t-1}$ denotes the longitudinal movement in meters of the target vehicle in Frenet coordinates of the target trajectory between two consecutive time steps. If the agent drives in the wrong way then the displacement reward will be multiplied by $-1$. The displacement reward provides a **dense reward** to encourage the agent to move forward. We set $c_{disp} = 1$.

2. The speed reward: $R_{speed} = v_t/v_{max}$, where $v_t, v_{max}$ denotes current speed and maximum allow speed in current road in $km/h$, respectively. If the agent drives in wrong way then the speed reward will be multiplied by $-1$. We set $c_{speed} = 0.1$.

3. The collision reward: $R_{collision} = 1$ if a collision with a vehicle, human, or object happens. Otherwise, it is $0$. The coefficient $c_{collision} = 5$.

4. The terminal reward: $R_{term}$ is non-zero only at the last time step. At that step, we set $R_{disp} = R_{speed} = R_{collision} = 0$ and assign $R_{term}$ according to the terminal state. $R_{term}$ is set to $+10$ if the vehicle reaches the destination (successes) and $-5$ if the vehicle drives out of the road.

For measuring the safety, collision to vehicles, obstacles, sidewalk raises a cost $+1$ at each time step. The sum of cost generated in one episode is the episodic cost.

In CARLA, we train and test agents in the Town01 environment. There exist many predefined routes in the town with different spawn locations and destinations. The length and spawn point of each route is randomized for each episode. We use the bird-eye view image in CARLA as observation and thus CNN is used as the feature extractor. In CARLA environment the reward function follows the reward function in MetaDrive safety benchmark.

In GTA V, we manually pick start and end coordinates in the world map to form multiple routes in different scenes. We split those scenes to the training and test sets. The training set contains two scenes with straight roads, turns, and medium traffic. The test set contains one different scene. For each episode, the traffic condition is randomized by the game engine. In GTA V environment the reward function follows the reward function in MetaDrive safety benchmark. The terminations include arriving at the destination (success), crashing with objects or vehicles for more than five frames (failure), and timeout (failure). The discrete keyboard input will be translated into continuous steering and acceleration signals for controls. Our customized human-in-the-loop compatible policy learning environment builds upon GTA V with a full set of well-defined observation, reward and termination conditions. The environment will be open-sourced and available to the community [1].

For these tasks, the reward function is composed of two parts: a sparse termination reward (+10 when reaching the destination) and a dense moving reward (the distance moving toward the destination within one step).

**Discrete action space environment.** For discrete action space, we test on MiniGrid Two Room task [4]. MiniGrid Two Room is a task requiring heavy exploration since the agent needs to move toward a door and open the door before reaching the destination. The spawn locations, the destinations, door locations and the geometry of each room are randomized. The observation of MiniGrid is the semantic map of agent's local neighborhood. MiniGrid tasks only support using the keyboard as the input device. Only in the MiniGrid task, we render the agent's action in the environment so that the human can decide whether to take over or return back based on both the current state and agent's action. But this is not feasible in other tasks since other tasks require real-time responses from humans and there is not enough time for humans to observe agent's actions even if we plot those actions in the visualization interface. Following the default reward function as in original repository, in MiniGrid environment a sparse reward is used. When the agent reach the goal, $+1$ reward is given and otherwise the reward is always $0$.

**Real-time experiment.** Note that the local computer we use for human-in-the-loop experiments, which has an Nvidia GeForce RTX 3080 gpu, can support real-time simulation and training. In MetaDrive and CARLA, the physics simulation is run at 10Hz in the virtual world and in GTA V the frequency is 30Hz. That is, each environmental step will cause the virtual world advancing 0.1 / 0.033 seconds. After each environment interaction, PVP updates its policy once, inferring and back-propagating one SGD batch. Our experience suggests that the local computer can effortlessly support concurrent running of the simulation with human-agent shared control as well as the background policy update at the frequency in wall-time higher than the simulation frequency in the virtual world. That is, our training can run at frequency higher than 10Hz / 30Hz so that the time-elapse is actually faster in the real world than in the virtual world. We will limit the FPS to the system frequency so that the human subjects experience realistic time-elapse.

**Control devices.** CARLA tasks use a Wheel, GTA V tasks use a keyboard, and MiniGrid tasks use a keyboard (provide discrete control signals). In all devices, a button is configured to indicate intervention. There is another button in the devices that activates an emergency stop. If any discomfort happens, human subjects can pause or stop the experiment immediately.

---

[1] The customized environment builds upon prior efforts on the communication between GTA V engine and Python interface: `https://github.com/aitorzip/DeepGTAV`, `https://github.com/gdpinchina/DeeperGTAV`, `https://github.com/aitorzip/VPilot`

# F  Extra Experimental Results

## F.1  Impact of Control Devices

Table 6: The impact of different human input devices in MetaDrive benchmark.

| Input Device | Method | Training | | | Testing | | |
|---|---|---|---|---|---|---|---|
| | | Human Data Usage | Total Data Usage | Total Safety Cost | Episodic Return | Episodic Safety Cost | Success Rate |
| Wheel | HACO | 21.2K (0.53) | 40K | 42 | 250.039 | 1.453 | 0.355 |
| | PVP | 10.3K (0.26) | 40K | 12 | 336.657 | 1.543 | 0.808 |
| Gamepad | HACO | 28.4K (0.71) | 40K | 55 | 71.37 | 1.97 | 0.0 |
| | PVP | 7.4K (0.19) | 40K | 21 | 356.99 | 1.31 | 0.920 |
| Keyboard | HACO | 19.2K (0.48) | 40K | 130 | 143.28 | 1.645 | 0.139 |
| | PVP | 14.6K (0.37) | 40K | 76 | 353.636 | 0.898 | 0.857 |

Table 6 presents the experiment results with different input devices in MetaDrive benchmark. In all settings, PVP agents outperform baseline method, showing the generalizability of PVP on different control devices. We observe that HACO [26] has performance discrepancy with different input devices. When using Gamepad, human subjects tend to push and pull the stick to the limits, producing extreme values. Extreme actions are particularly harmful to previous method as it does not incorporate the regularization terms on Q function to bound the Q values. When using the keyboard, the human subjects press arrow keys to indicate increasing/decreasing current steering/acceleration values for an increment. Therefore there will be fewer extreme values happening than using a Gamepad, which explains why the baseline HACO performs better with the keyboard compared to Gamepad. Due to less extreme values, when using Steering Wheel, HACO achieves good performance.

**A case study in a toy environment.** We retrieve the agents stored during the training of HACO and PVP in CARLA task. We test them in a straight road in CARLA town and plot their actuating signals in Fig. 9. In this task, the steering should be always close to zero. However, we find that as the training iterations increase, the HACO agents gradually demonstrate unstable steering and their steering is deviating. In human-AI shared control, such unstable behaviors force human subjects to involve frequently. In contrast, the PVP policies learn a much better solution in lane keeping.

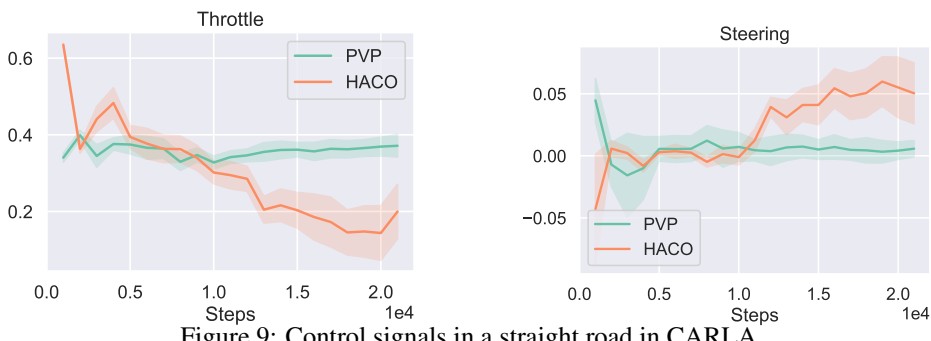

Figure 9: Control signals in a straight road in CARLA.

**Visualization of action sequences in training.** In Fig. 10, we present the visualization of the trajectories during human-robot shared control. Comparing the visualization of HACO and PVP, we find that PVP generates smoother trajectories. Stable and smooth agent actions greatly improve human subjects' experience and relieve their stress during human-robot shared control. We can also find that as the training goes, PVP requires less human involvement. These results explain the performance of PVP and is aligned with the behavior shown in the supplementary video.

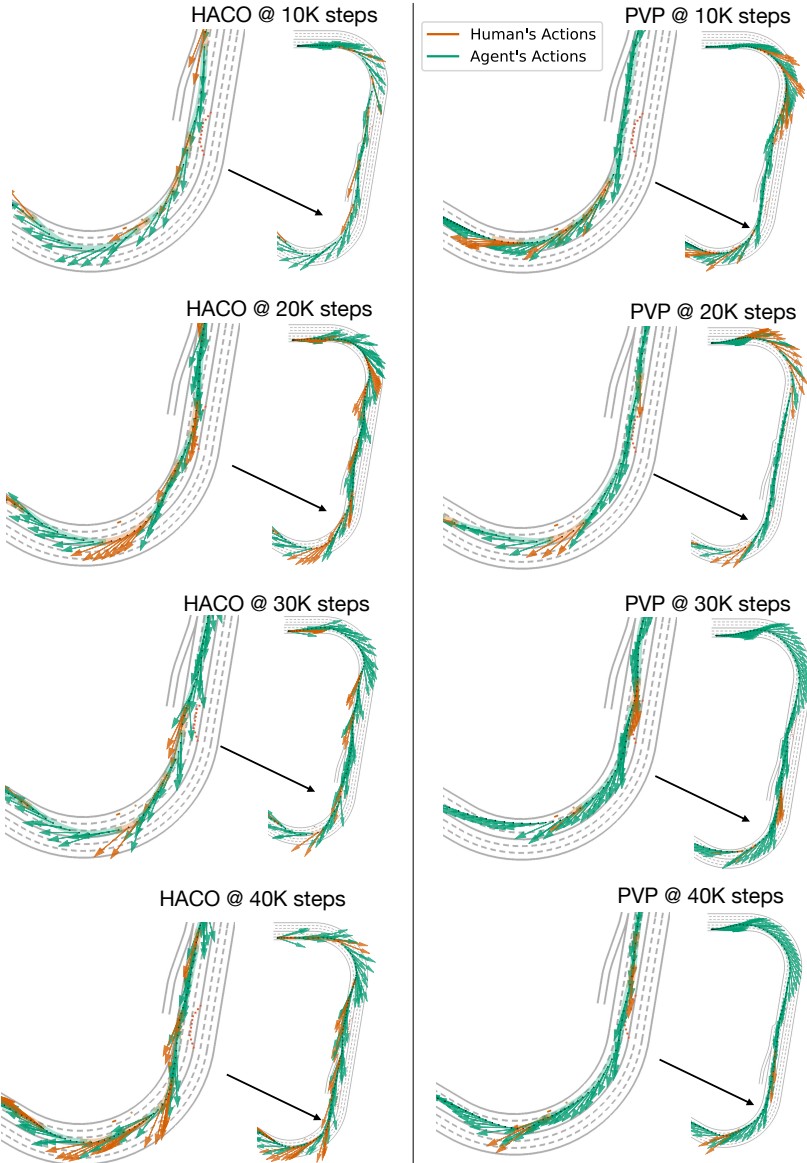

Figure 10: In MetaDrive task, we use the top-down view to plot the trajectories of human-agent shared control. We use dense arrows to represent the actions that are applied to the environments. The arrow starts at the position of the car at that time step and its direction is the steering angle, projected into ego car's local coordination. The length of the arrow represents the acceleration. We use green and yellow arrows to denote agent's actions and human's actions, respectively.

## F.2    Extra Results in MiniGrid

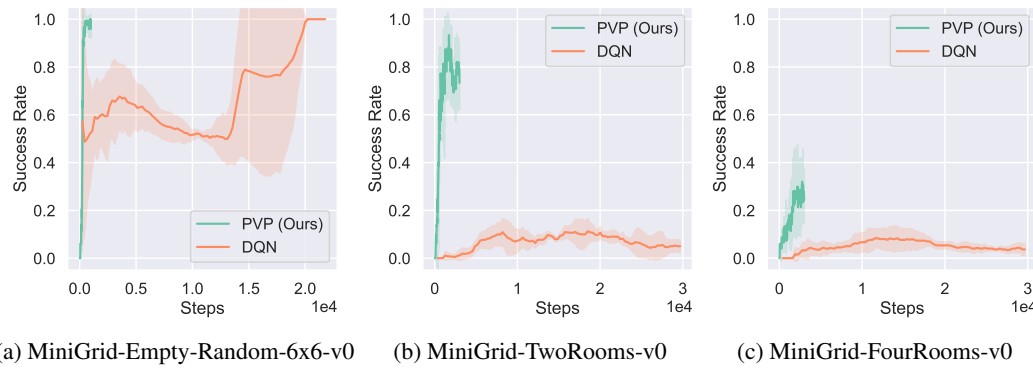

(a) MiniGrid-Empty-Random-6x6-v0    (b) MiniGrid-TwoRooms-v0    (c) MiniGrid-FourRooms-v0

Figure 11: MiniGrid results.

In Fig. 11, we present the extra results in two additional MiniGrid environments. PVP achieves superior performance compared to RL baseline. Note that we use a CNN without recurrent module as the feature extractor. The performance of PVP can be further improved if we utilize the neural architecture with memory capability.

# G    Hyper-parameters

In MetaDrive safety benchmark [25] task, the observation is a state vector. There exists a split of training and test environments in MetaDrive. We present the result of the learned agent performing in the test environment.

In CARLA [8], the observation is the bird-eye view image in $[84, 84, 5]$ shape, where 5 is the number of semantic channels. We train and evaluate the agents in the same NoCrashTown01 environment.

In GTA V, the observation is a state vector containing 2D LiDAR scanning for 240 total sampling points with max range 50m [15], vehicle state variables (speed, throttle, steering, heading), and navigation state variables (distance to road borders, distance to the next navigation point, collision to objects). There exists a split of training and test environments in GTA V. We present the result of the learned agent performing in the test environment.

In MiniGrid tasks [4] MiniGrid-Empty-Random-6x6-v0 (Empty Room), MiniGrid-MultiRoom-N2-S4-v0 (Two Room) and MiniGrid-MultiRoom-N4-S5-v0 (Four Room), the observation is the top-down view semantic map in shape $[7, 7, 3]$.

In MetaDrive and GTA V, we use a MLP with two hidden layers, each has 256 units and ReLU activation, as the network architecture for the value network and policy network.

For CARLA task, since the input image has the same size of [84, 84] pixels, we use the same 5-layers CNN architecture with [16, 32, 64, 128, 256] filters in each layer. The corresponding kernel-size is [[4, 4], [3, 3], [3, 3], [3, 3], [4, 4]], and strides [3, 2, 2, 2, 4]. We use ReLU as activation functions between each layer.

For MiniGrid tasks, we use a 3-layer CNN architecture with [16, 16, 32] filters in each layer. All three layers have kernel-size 2 and there is a max-pooling layer between the first two layers. We use ReLU as activation functions between each layer.

### Table 7: PVP (MetaDrive)

| Hyper-parameter | Value |
| --- | --- |
| Discounted Factor $\gamma$ | 0.99 |
| $\tau$ for Target Network Update | 0.005 |
| Learning Rate | 0.0001 |
| Steps before Learning Start | 100 |
| Steps per Iteration | 1 |
| Gradient Steps per Iteration | 1 |
| Train Batch Size | 100 |
| Q Value Bound | 1 |

### Table 8: PVP (CARLA)

| Hyper-parameter | Value |
| --- | --- |
| Discounted Factor $\gamma$ | 0.99 |
| $\tau$ for Target Network Update | 0.005 |
| Learning Rate | 0.0001 |
| Steps before Learning Start | 100 |
| Steps per Iteration | 1 |
| Gradient Steps per Iteration | 1 |
| Train Batch Size | 128 |
| Q Value Bound | 1 |

### Table 9: PVP (GTA V)

| Hyper-parameter | Value |
| --- | --- |
| Discounted Factor $\gamma$ | 0.99 |
| $\tau$ for Target Network Update | 0.005 |
| Learning Rate | 0.0001 |
| Steps before Learning Start | 100 |
| Steps per Iteration | 1 |
| Gradient Steps per Iteration | 1 |
| Train Batch Size | 100 |
| Q Value Bound | 1 |

### Table 10: PVP (MiniGrid)

| Hyper-parameter | Value |
| --- | --- |
| Discounted Factor $\gamma$ | 0.99 |
| $\tau$ for Target Network Update | 0.005 |
| Learning Rate | 0.0001 |
| Steps before Learning Start | 50 |
| Steps per Iteration | 1 |
| Gradient Steps per Iteration | 32 |
| Target Network Update Interval | 1 |
| Train Batch Size | 256 |
| Q Value Bound | 1 |
| Exploration Reducing Fraction | 0 |
| Random Action Probability Initial Value | 0 |
| Random Action Probability Final Value | 0 |

### Table 11: HACO (MetaDrive)

| Hyper-parameter | Value |
| --- | --- |
| Discounted Factor $\gamma$ | 0.99 |
| $\tau$ for Target Network Update | 0.005 |
| Learning Rate Actor | 0.0003 |
| Learning Rate Critic | 0.0003 |
| Learning Rate Entropy | 0.0003 |
| Steps before Learning Start | 100 |
| Steps per Iteration | 1 |
| Gradient Steps per Iteration | 1 |
| Target Network Update Interval | 1 |
| Train Batch Size | 128 |
| CQL Loss Temperature | 1.0 |

### Table 12: HACO (Carla)

| Hyper-parameter | Value |
| --- | --- |
| Discounted Factor $\gamma$ | 0.99 |
| $\tau$ for Target Network Update | 0.005 |
| Learning Rate Actor | 0.0003 |
| Learning Rate Critic | 0.0003 |
| Learning Rate Entropy | 0.0003 |
| Steps before Learning Start | 100 |
| Steps per Iteration | 1 |
| Gradient Steps per Iteration | 1 |
| Target Network Update Interval | 1 |
| Train Batch Size | 128 |
| CQL Loss Temperature | 1.0 |

### Table 13: TD3 (MetaDrive)

| Hyper-parameter | Value |
| --- | --- |
| Discounted Factor $\gamma$ | 0.99 |
| $\tau$ for Target Network Update | 0.005 |
| Learning Rate | 0.0001 |
| Steps before Learning Start | 10000 |
| Steps per Iteration | 1 |
| Gradient Steps per Iteration | 1 |
| Train Batch Size | 100 |

### Table 14: TD3 (Carla)

| Hyper-parameter | Value |
| --- | --- |
| Discounted Factor $\gamma$ | 0.99 |
| $\tau$ for Target Network Update | 0.005 |
| Learning Rate | 0.0001 |
| Steps before Learning Start | 10000 |
| Steps per Iteration | 1 |
| Gradient Steps per Iteration | 1 |
| Train Batch Size | 100 |

Table 15: TD3 (GTA V)

| Hyper-parameter | Value |
| --- | --- |
| Discounted Factor $\gamma$ | 0.99 |
| $\tau$ for Target Network Update | 0.005 |
| Learning Rate | 0.0001 |
| Steps before Learning Start | 10000 |
| Steps per Iteration | 1 |
| Gradient Steps per Iteration | 1 |
| Train Batch Size | 100 |

Table 16: DQN (MiniGrid)

| Hyper-parameter | Value |
| --- | --- |
| Discounted Factor $\gamma$ | 0.99 |
| $\tau$ for Target Network Update | 0.005 |
| Learning Rate | 0.0001 |
| Steps before Learning Start | 50 |
| Steps per Iteration | 1 |
| Gradient Steps per Iteration | 32 |
| Target Network Update Interval | 1 |
| Train Batch Size | 256 |
| Exploration Reducing Fraction | 0.3 |
| Random Action Probability Initial Value | 0 |
| Random Action Probability Final Value | 0.05 |

