# OpenReview forum: "Learning from Active Human Involvement through Proxy Value Propagation"
_NeurIPS.cc/2023/Conference — NeurIPS 2023 spotlight_

### Official Review · Reviewer_sDoy · 2023-06-25

**Soundness:** 4 excellent
**Presentation:** 3 good
**Contribution:** 4 excellent
**Rating:** 8
**Confidence:** 5

**Summary:**

The work introduces Proxy Value Propagation (PVP), an approach to reinforcement-learning without ground-truth rewards by instead using human intervention as a signal of quality. The value of a state-action pair is determined by a human intervention, where interventions signal that the agent's behavior was bad (and we assume that the human's action was good). Compared to naive RL approaches or other human-intervention methods, such as HG-DAgger, PVP learns self-driving policies much faster and with fewer crashes than all other approaches. The work is also tested with a user-study, gathering demonstration data from real humans.

**Strengths:**

Technical contribution:
* The proposed method works very well empirically, outperforming baselines with and without human intervention.
* The relative contribution of different components of the PVP approach are tested with an ablation study in the CARLA domain.
* Experiments cover several domains including continuous/discrete actions, different input mechanisms, and different state observations.
* The proposed approach specifically considers balance between human/agent labels, making sure to consider human preferences even when there are relatively few demonstrations/corrections.

Experiments:
* The experiments are clearly described and appropriately test the proposed PVP method against baselines.
* Human users are recruited and the human-intervention method is tested with real users, not synthetic/heuristic labeling.

Novelty:
* The paper compares well against prior work and the related works section covers most important previous approaches to the problem.
* The experiments compare against several baselines in prior work.

Clarity:
* The method is intuitive and easy to follow. Figures and tables clearly communicate the results.
* The supplementary material provides additional detail where useful (hyper-parameters, control schemes, etc.).

Reproducibility:
* Code is provided to reproduce all experiments.

**Weaknesses:**

Technical contribution:
* Prior work in human-labeling and intervention has found that human labelers are slow to react, and this time-delay must be accounted for (e.g. in [1], the authors needed to use eligibility traces to account for _which_ agent action was incorrect). Strange that this did not surface as a problem in these studies.
* The proposed treatment of agent vs. human Q-values could result is serious policy degradation if the human labeler makes mistakes or if the demonstrator is sub-optimal.
* Fusion of PVP with environment reward resulted in _worse_ performance.
* Removing epsilon greedy exploration from DQN effectively forces humans to do the exploration for PVP, which seems to work well in the driving simulator, but could be an issue in other domains.

User study:
* Information on the study is lacking. How many participants? What about background in computer science or driving? Mean age? What sorts of instructions were provided? Were participants allowed to practice first? These details can be in the supplement if they are not critical to the analysis of the algorithm, but they should be included for a user study.

Novelty:
* The approach seems very similar to [1], which treats human labels as indications of advantage instead of simply as reward (as in DeepTAMER, which is cited in the paper). While comparison to [1] is perhaps not necessary (as the paper compares thoroughly to several state-of-the-art approaches), [1] should at least be mentioned.
* Should be aware of other approaches to human-intervention labeling (e.g., [2], which also considered driving feedback, or [3] which performs BC and then RL)


[1] James MacGlashan, Mark K Ho, Robert Loftin, Bei Peng, Guan Wang, David L. Roberts, Matthew E. Taylor, and Michael L. Littman. 2017. Interactive learning from policy-dependent human feedback. In Proceedings of the 34th International Conference on Machine Learning - Volume 70 (ICML'17). JMLR.org, 2285–2294.
[2] Mariah L. Schrum, Erin Hedlund-Botti, Nina Moorman, and Matthew C. Gombolay. 2022. MIND MELD: Personalized Meta-Learning for Robot-Centric Imitation Learning. In Proceedings of the 2022 ACM/IEEE International Conference on Human-Robot Interaction (HRI '22). IEEE Press, 157–165.
[3] Cheng, Ching-An, Xinyan Yan, Nolan Wagener and Byron Boots. “Fast Policy Learning through Imitation and Reinforcement.” Conference on Uncertainty in Artificial Intelligence (2018).

**Questions:**

1. Is there any drawback to the proposed approach to action labeling when compared to something like COACH [1]? Should PVP be considering the time-delay of human feedback?
2. Line 187-188 states that the novice-policy (agent behavior) contains information of the forward dynamics. What does this mean?
3. The addition to the CQL loss is likened to an L2 regularization on the Q-values from the agent, but later experiments adding this L2 penalty to CQL seem to perform much worse than PVP. Why is there a discrepancy between the two? Shouldn't CQL + L2 == PVP, as in Equation 6?
4. It seems that PVP relies heavily on humans doing the initial exploration. Would something like LOKI [3] perform comparably, or is the value propagation part of the key to success?
5. Please address the questions listed above concerning the user study. Additional detail should be provided.

**Limitations:**

The authors list and acknowledge several limitations. Mitigations to these are left to be the subject of future work.

Societal impact of the work is not explicitly addressed, though an ethics statement does appropriately address IRB for the study.

---

> ### Author Rebuttal · Authors · 2023-08-09
>
> Thank you for your review!
>
> ---
> > **W1:** *Prior work in human-labeling and intervention has found that human labelers are slow to react, and this time-delay must be accounted for. Strange that this did not surface as a problem in these studies.*
>
> **Re:**
>
> Indeed, we do not consider the time-delay in the current method. The problem is mitigated as PVP learns from active human involvement. If the agent learns a time-delayed policy, then human subjects will correct such behaviors in later training.
>
> The eligibility traces in [1] allow feedback to affect earlier decisions. The TD loss in our system serves a similar role that can propagate proxy values (considering it as an implicit form of human feedback, as it is not hard-coded to $\pm 1$ for each action) to previous steps.
>
> ---
> > **W2:** *The proposed treatment of agent vs. human Q-values could result in serious policy degradation if the human labeler makes mistakes or if the demonstrator is sub-optimal.*
>
> **Re:**
>
> In our empirical studies, we found that mistakes made by the human labeler can indeed impact the policy during the early stages of training. However, this degradation is usually smoothed out as training progresses, with correct human demonstrations mitigating the impact of early errors.
>
> Due to resource limits, we did not perform quantitative analysis on how the level of noise (or how the quality of demonstration) affects policy learning. As we shown in the analysis in Appendix D Theorem D.3, the training-time performance is bounded by the human’s performance in providing action and in intervention. The quality of demonstration and intervention does affect the performance of PVP.
>
> ---
> > **W3:** *Fusion of PVP with environment reward resulted in worse performance.*
>
> **Re:**
>
> As presented in the one-page PDF, the reward function we used for driving tasks has 4 terms. The displacement reward, the speed reward, the collision penalty and the termination reward. However, as we discussed in the experiment section, the environmental reward might not necessarily align with human preference. Human subjects usually demonstrated more fine grained behaviors such as decelerating before crossroad or maintaining a safe distance from the front vehicles. Those behaviors might contradict the speed reward as it encourages high speed. Such misalignment between reward functions and human preference leads to worse performance.
>
> ---
> > **W4:** *Information on the human subject study is lacking.*
>
> **Re:**
>
> Thank you for the advice! Section 1 in the one-page PDF discusses the requirements for human subjects, the onboarding period and the information / advice we provide to the human subjects.
>
> We summarize the answers to your concerns:
> * We recruit 5 human subjects.
> * All of them are college students with an average age of 21.4 years old.
> * They go through an onboarding procedure before the main experiments when they can practice the tasks with different control devices.
> * The participants are informed by the objectives: driving safely to the destination and ensuring the behavior aligns with traffic regulations and human preferences.
> * The participants are encouraged to perform intervention whenever they perceive the vehicle might be in a dangerous situation, in violation of traffic rules, or in whatever scenario the human subjects feel they wouldn't behave in the way the novice policies do.
>
> ---
> > **W5:** *The approach seems very similar to [1]. [1] should at least be mentioned. Should be aware of other approaches to human-intervention labeling.*
>
> **Re:**
>
> Thank you for bringing the insightful works [1, 2, 3]. We will discuss these papers in the revision. Compared to COACH [1], PVP accepts not only the feedback (the intervention signal) but also the human demonstration. Besides, PVP uses TD loss to build a proxy value function instead of using the feedback directly to train the policy. However, PVP does not consider the time-delay of human subjects explicitly as COACH.
>
> ---
> > **Q1:** *Is there any drawback when compared to something like COACH [1]? Should PVP be considering the time-delay of human feedback?*
>
> **Re:**
>
> We do not consider the time-delay in the current method explicitly and this is indeed a limitation.
>
> However, PVP has two features that can mitigate this issue. The eligibility traces in COACH allow feedback to affect earlier decisions. The TD loss in our system serves a similar role that can propagate proxy values (considering it as an implicit form of human feedback, as it is not hard-coded to $\pm 1$ for each action) to previous steps.
>
> Meanwhile, PVP trains a deterministic novice policy and thus its behavior is consistent in a short period of time. Human subjects can relatively easier to determine the intention of the novice policy and thus make an intervention accordingly, compared to DAgger or HACO where there exists stochasticity in the novice policy. Therefore the time-delay of human feedback might not be a large issue.
>
> Last but not least, as you can see in the human subject research protocol, the objective of the driving task we told the human subjects is to arrive at the destination safely while the behavior should be human-like. We don't require the human subjects to drive as fast as possible. Thus the requirement of real-time response is relaxed.
>
> ---
> > **Q2:** *The addition to the CQL loss is likened to an L2 regularization on the Q-values from the agent, but experiments adding L2 penalty to CQL seem to perform much worse than PVP. Why is there a discrepancy between the two? Shouldn't CQL + L2 == PVP, as in Equation 6?*
>
> **Re:**
>
> There might be a misunderstanding. In the ablation study section (Line 339 and Table 4), we conduct experiments to evaluate the performance of the pure CQL objective (without regularization) while other PVP designs remain the same. This experiment shows that pure CQL is not applicable to this human active involvement setting. We will modify the wording to address the misunderstanding.

---

> > ### Comment · Reviewer_sDoy · 2023-08-11
> >
> > Thanks to the authors for their time and their thorough response.
> >
> > The TD loss could serve as a way of "backing-up" value estimates given that the agent uses a deterministic driving policy, but this assumes that the deterministic policy is consistent through time and not sensitive to subtle variations in the input state (which is not guaranteed, particularly as the agent moves to new maps). Further, it is likely that interventions will occur in dynamic/unpredictable states (e.g., during a lane-merge, a turn, or a changing traffic light), and a naive back-up of value might lead to undesirable behavior in these states (e.g., assigning higher value to acceleration while waiting a red light).

---

> > ### Comment · Reviewer_e73G · 2023-08-21
> > **Acknowledgement of rebuttal**
> >
> > Thanks for the detailed response and clarifying any misunderstandings I may have had. I also read through the responses to other reviews. I still think this is a very strong paper and will keep my score at Strong Accept.

---

### Official Review · Reviewer_iaMb · 2023-07-04

**Soundness:** 3 good
**Presentation:** 4 excellent
**Contribution:** 3 good
**Rating:** 7
**Confidence:** 4

**Summary:**

The paper proposes an approach for learning from human interventions based on offline RL with pseudo-rewards. The paper assumes a shared autonomy setting where a human monitors an agents rollouts and can intervene whenever deemed necessary and provide corrective demonstrations. The algorithm relabels those interventions with positive Q value targets and the faulty policy actions that led to intervention with negative Q targets. It then trains the policy on these pseudo-rewards with offline RL. In extensive experiments on multiple driving simulators with differing human input devices, the approach leads to faster and more successful learning of desired behaviors than prior RL and learning-from-human-intervention approaches.

**Strengths:**

- the paper's approach is elegant and simple, it leverages common offline RL techniques in combination with pseudo-reward labels and, due to it's simplicity, seems easily applicable to other domains

- the empirical evaluation of the approach is comprehensive, with numerous baselines spanning the relevant related works, multiple environments and many meaningful ablations

- the paper is well-written and easy to understand. it has intuitive visualizations for illustrating the approach, evaluation environments and qualitative results.



**Weaknesses:**

(A) **Concerns about training objective**: the training objective sets the **Q-value targets** of the human intervention trajectories to +1 and the Q-value targets of the policy actions in those intervention trajectories to -1. I have a few concerns about this objective:

  (1) by setting the Q-value target instead of setting the reward, the objective does not allow reward propagation through these updates, e.g. from post-intervention states. At the same time the algorithm runs "regular" Q-updates with full propagation and adds both objectives. Why not set **rewards** instead of Q-targets to indicate good and bad behavior and only run the regular Q-update on all data samples? That seems more elegant than the current two-part objective

  (2) the current objective assigns low Q-value to all policy actions during human intervention. However, the human intervention only provides a signal that the sequence of policy actions **before** the intervention were bad. It would for example be possible that the human "over-intervenes" and keeps providing demonstrations in states in which the policy's actions are already good again. In these cases the current objective would penalize good policy actions

(B) **Only navigation environments**: while the experimental evaluations are comprehensive, they are limited to navigation environments (driving and grid world navigation). It would strengthen the paper to show the same algorithm can work for teaching robot manipulation tasks, e.g. by performing teleoperation in a simulated robotic environment.

(C) **Unclear human user experiments**: all experimental evaluations are performed with real humans in the pipeline. While this is generally positive, it means special care needs to be taken to ensure fair comparability between all methods, since different users may interact with systems differently, they learn and adapt to systems over time etc. Thus, human user studies require great care in execution and documentation, eg ensuring that the same users interact with all methods and baselines (to account for differences between individual users), the order of methods is randomized and all users are given some time to practice with the system (to account for adaptation effects) etc. The paper does not provide any details in this regard for the quantitative evaluations in Tables 1, 2 and 4. It would be important to provide more details on the design of all experiments involving human users to ensure fair comparison.



**Questions:**

(Q1): How could the proposed method be combined with pre-trained policies or ground truth rewards?
--> the current method requires lots of human teaching, it would be beneficial if human teaching was only used to *supplement* autonomous agent learning from some external source of supervision. The one experiment in the paper that does combine it with external rewards seems unsuccessful -- how can this be fixed?

(Q2): How can the proposed method be extended to situations where humans cannot directly provide demonstrations?
--> eg for an ant robot it is impossible for humans to provide demonstrations (the authors mention this short-coming of the current approach in the limitations section). I am wondering whether you have any ideas how this could be extended to situations like the ant robot?

**Limitations:**

- add more details about human experiment design

- add a (simulated) robotics manipulation environment

- clarify concerns about the proposed objective


## Summary of Review

Overall, I think this is a strong paper. It combines a novel, yet simple and elegant approach with an extensive evaluation and comparison to numerous prior works. While I would like the authors to address the points raised in my review, I am happy to accept the submission.

---

> ### Author Rebuttal · Authors · 2023-08-09
>
> Thank you for your review! Please find our responses as follows.
>
> ---
>
> ### Weaknesses
>
> > **(A1)** *Why not set rewards instead of Q-targets?*
>
> Please refer to the response to W1 of Reviewer pvGG.
>
> > **(A2)** *The current objective assigns low Q-value to all policy actions during human intervention. However, human intervention only provides a signal that the sequence of policy actions before the intervention were bad. It would be possible that the human "over-intervenes" and keeps providing demonstrations in states the policy's actions are already good again. In these cases the current objective would penalize good policy actions.*
>
> **Response:**
>
> Let's suppose the "over-intervention" happens and this means the human's action is identical to the agent's action $a_n = a_h$. At this step, PVP will try to assign high proxy value to $Q(s, a_h)$ (regressing to $+1$) and low proxy value to $Q(s, a_n)$ (regressing to $-1$). However because $a_h = a_n$, their proxy values will cancel each other:
>
> $
> \min (Q(s, a_h) - 1 )^2 + (Q(s, a_n) + 1)^2 + \text{TD Loss}
> \equiv
> \min 2Q^2(s, a) + \text{TD Loss}
>  $
>
> Under this circumstance, the proxy value still prefers this action because those human-intervening actions have much lower value (near $-1$) while those good actions have values close to 0.
>
> ---
>
> > *(B) Only navigation environments: while the experimental evaluations are comprehensive, they are limited to navigation environments (driving and grid world navigation). It would strengthen the paper to show the same algorithm can work for teaching robot manipulation tasks, e.g. by performing teleoperation in a simulated robotic environment.*
>
> **Response:**
>
> We totally agree! It is a promising future direction to apply PVP in robot manipulation tasks. There is a huge challenge to existing RL algorithms in robot manipulation tasks since fine manipulation tasks such as threading a zip tie or juggling a ping pong ball is very hard to be characterized by a reward function without sufficient engineering efforts. The proposed PVP method can learn from human demonstration behaviors in an online manner and provide corrective feedback to the agent. Experimenting on Robot manipulation tasks will be our immediate future extension. We plan to utilize the teleoperation hardware developed in a recent work [A] to conduct the real-world experiment.
>
>
> [A] Learning Fine-Grained Bimanual Manipulation with Low-Cost Hardware. RSS’23
>
> ---
>
> > *(C) Unclear human user experiments: all experimental evaluations are performed with real humans in the pipeline. While this is generally positive, it means special care needs to be taken to ensure fair comparability between all methods. Thus, human user studies require great care in execution and documentation, eg ensuring that the same users interact with all methods and baselines (to account for differences between individual users), the order of methods is randomized and all users are given some time to practice with the system (to account for adaptation effects) etc. It would be important to provide more details on the design of all experiments involving human users to ensure fair comparison.*
>
> **Response:**
>
> Thank you for your advice!
>
> We provide a one-page PDF on human subject research protocol in the common rebuttal, where we discuss the requirement to human subjects, the onboarding period and the information / advice we provide to the human subjects.
>
> For your concern on the proficiency biases, we do randomize the order of experiments with different control devices, tasks and training algorithms for each participant. Therefore we can average out the "adaptation effects".
>
> ---
> ### Questions
>
> > *(Q1): How could the proposed method be combined with pre-trained policies or ground truth rewards? --> the current method requires lots of human teaching, it would be beneficial if human teaching was only used to supplement autonomous agent learning from some external source of supervision. The one experiment in the paper that does combine it with external rewards seems unsuccessful -- how can this be fixed?*
>
> **Response:**
>
> With the recent successes on large decision models such as Robotic Transformer 2 (RT-2), pretraining policy on large datasets is a promising approach to further improve the proposed method.
>
> We haven't conducted an experiment on this. We have an ablation study that uses both the environmental reward and human demonstration at the same time. Because two supervision signals are provided at the same time, they might not be fully aligned and this causes confusion to the model. For example, humans might want to slow down before a crossroad, while the speed reward in our reward function encourages the agent to move as fast as they can.
>
> In future work, we can experiment with a natural idea: the two-stage training scheme. We can first pretrain policy with ground truth reward or imitation learning and in the second stage we can incorporate the proposed PVP method to finetune the policy to improve its preference alignment such as improving the safety-critical performance.
>
> ---
>
> > *(Q2): How can the proposed method be extended to situations where humans cannot directly provide demonstrations? --> e.g. for an ant robot it is impossible for humans to provide demonstrations (the authors mention this short-coming of the current approach in the limitations section). I am wondering whether you have any ideas how this could be extended to situations like the ant robot?*
>
> **Response:**
>
> We can learn a goal-conditioned policy first, which also connects to your first idea using pretraining. We will then conduct the human-AI shared control in the second stage: we can invite human subjects to give demonstrations on the goal space, instead of the low-level action space. PVP can thus learn a high-level control policy from human demonstration and intervention that produces high-level goals for the low-level controller.

---

> > ### Comment · Reviewer_iaMb · 2023-08-16
> > **Thank you for your rebuttal!**
> >
> > Thank you for your response. I am generally happy with how it addresses my comments. I appreciate the explanations for my concern (A2) and the inclusion of details regarding the human user studies!
> >
> > I am still a bit unsure about question (A1), ie reward vs value targets. How is this different from off-policy RL where we need to learn from transitions collected with a different policy?
> >
> > Additionally, the failure case mentioned seems like it could be avoided by giving some strong penalty for human involvement that outweighs the benefits of the following good actions? This would encourage the policy to avoid interventions but still follow the good actions after?
> >
> > I would appreciate the authors insight on this question, but am in general happy to recommend acceptance for the submission.

---

### Official Review · Reviewer_pvGG · 2023-07-09

**Soundness:** 3 good
**Presentation:** 3 good
**Contribution:** 3 good
**Rating:** 5
**Confidence:** 3

**Summary:**

This paper focuses on human-in-the-loop for reward-free policy learning, wherein a human has the option to override the policy by taking over control when the agent attempts to perform risk behaviors. To solve this problem, this work presents Proxy Value Propagation (PVP), which directly assigns positive values (+1) to the actions induced by the human interruption and negative values (-1) to the actions the agent takes in specific states. By manipulating the values, PVP encourages the agent to approximate human behaviors while avoiding the actions intervened by human.


**Strengths:**


- The problem was well-motivated and the setting is realistic.

- The empirical results suggest PVP outperforms a set of baselines involving imitation learning and other human-in-the-loop methods in terms of multiple measurements. Furthermore, the experiments with human subjects are convincing and extensive.

- The proposed method is simple and efficient.


**Weaknesses:**

One major concern is that the technique of directly manipulating the value seems to be not well-motivated. As claimed in lines 178-180, the optimal policy should approximate the behaviors of human subjects while avoiding performing the intervened actions. Starting from this motivation, two natural ways are 1) assigning high rewards to human actions and low rewards to intervened policy actions, and 2) constraining the policy by manipulating the likelihood of the specific actions. Since the value function induced by TD learning measures the long-term performances starting from specific state-action pairs, directly assigning +1 or -1 to the value function seems to tell the agent that it tends to succeed or fail starting from the s-a pairs. However, human might only take control at some discrete time steps, and the subsequent behaviors after the interruption are unpredictable. Thus, the natural design is to assign manufactured rewards (+1/-1) to the specific transitions from my perspective. Though I believe that directly manipulating the value and propagating through TD-learning might be a better choice, it is worth comparing with the naive baseline mentioned above and further discussing the advantages of PVP.

Though learning with online human interruptions is quite an interesting setting, it is pretty ideal to me as online learning is risky and expensive even with a human guardian in several domains (e.g., robotics). In terms of the driving case investigated in the paper, I believe that offline learning with human interruption datasets might be more realistic and interesting given numerous in-the-wild datasets. However, I would agree the investigation is out of scope concerning the current paper.


**Questions:**

- If the reward function is set to zero, why $Q=\sum r_t$ can be set to $+1$ or $-1$?

- For two adjacent time steps in an episode, if both $Q(s_1,a_1)$ and $Q(s_2,a_2)$ are set to 1, how can it satisfy Bellman equation as $Q(s_1,a_1) = \gamma max_a Q(s_2,a)$?

---

> ### Author Rebuttal · Authors · 2023-08-09
>
>
> Thank you for your review! Please find our responses as follows.
>
>
> ---
>
> ### Weaknesses
>
> > **W1:** *One major concern is that the technique of directly manipulating the value seems to be not well-motivated. As claimed in lines 178-180, the optimal policy should approximate the behaviors of human subjects while avoiding performing the intervened actions. Starting from this motivation, two natural ways are 1) assigning high rewards to human actions and low rewards to intervening policy actions, and 2) constraining the policy by manipulating the likelihood of the specific actions. Since the value function induced by TD learning measures the long-term performances starting from specific state-action pairs, directly assigning +1 or -1 to the value function seems to tell the agent that it tends to succeed or fail starting from the s-a pairs. However, humans might only take control at some discrete time steps, and the subsequent behaviors after the interruption are unpredictable. Thus, the natural design is to assign manufactured rewards (+1/-1) to the specific transitions from my perspective. Though I believe that directly manipulating the value and propagating through TD-learning might be a better choice, it is worth comparing with the naive baseline mentioned above and further discussing the advantages of PVP.*
>
> **Response:**
>
> Though the idea of assigning reward +1 to human's actions and -1 to intervention actions is straightforward, it doesn't work in practice. This is because in the framework of value-based RL, the reward participates in the learning via the TD loss:
>
> $
> J^\text{TD}(\theta) = \mathbb E_{(s, a, s')} | Q_\theta(s, a) - (r(s, a) + \gamma \max_{a'} Q_{\hat{\theta}}(s', a')) |^2
> $
>
> You can see that the Bellman backup is conducted on the transition triplet $(s, a, s')$, where you have to use future's value with current reward as an estimator of current value. Therefore, the reward must correspond to the action from the behavior policy, the action $a$ causes the transitions from $s$ to $s'$. In our context, during human involvement, the action $a = a_h$ must come from human policy as the human subject is taking control $s' \sim \mathcal P(s, a_h)$. And thus the reward will be $+1$ for those human-involving transitions.
>
> Therefore, we have no way to "assign low rewards to intervening policy actions $a_n$" because we need to know the future state $s'' \sim \mathcal P(s, a_n)$ caused by this action so that we can compute the TD target. However, we can only have the future state caused by human action. It is also not practical to query the environment to get the next state $s''$ as the $a_n$ is potentially a danger / undesired action and there is no way to replay it in the environment.
>
> In our preliminary experiment, we have tried this idea and the policy failed to learn anything no matter how much human involvement is provided. The reason is the reward will always be $+1$ for the transitions that occur during human involvement. And the learning agent found a pitfall to maximize its reward: it always demonstrates undesired behaviors, e.g. making a sharp left turn toward the road edge, so that humans will always take control which causes +1 reward.
>
>
> ---
>
> > **W2:** *Though learning with online human interruptions is quite an interesting setting, it is pretty ideal to me as online learning is risky and expensive even with a human guardian in several domains (e.g., robotics). In terms of the driving case investigated in the paper, I believe that offline learning with human interruption datasets might be more realistic and interesting given numerous in-the-wild datasets. However, I would agree the investigation is out of scope concerning the current paper.*
>
> **Response:**
>
> We agree that offline learning is a very important setting that is worth a lot of effort. The active human involvement being tackled in this submission is an orthogonal research setting and has its unique value in safety critical systems. In the development and deployment of learning-based systems, such as self-driving cars, human-AI shared control is a common scenario as there is usually a human driver sitting behind the wheel preparing to take over the control. Our proposed method can effectively leverage the human data collected during such shared control.
>
> Offline learning can help augment PVP, especially when we want to scale up the proposed method. We can introduce a pretraining stage before the human-AI shared control in PVP. By doing so the pretrained base policy learns some basic skills from in-the-wild datasets to reduce the human cost while achieving human alignment in later stages when training with PVP.
>
> ---
>
> ### Questions
>
>
> > **Q1:** *If the reward function is set to zero, why $Q = \sum r_t$ can be set to +1 or -1? For two adjacent time steps in an episode, if both $Q(s_1, a_1)$ and $Q(s_2, a_2)$ are set to 1, how can it satisfy Bellman equation as $Q(s_1, a_1) = \gamma \max_a Q(s_2, a)$?*
>
> **Response:**
>
> The proxy value assignment is an auxiliary learning objective but not a hard constraint to the proxy value function. This is because apart from the PVP loss, we also have the TD loss and thus the proxy value might not regress to $\pm 1$.
>
> In the context of this work, the proxy value function is not an estimator of the discounted return, as we don't have a reward function at all. Instead, the proxy value function is a function measuring human preference and can be used to select human preferable actions when applying the argmax rule on it $a = \arg\max_a Q(s, a)$. That is to say, the proxy value function can induce desired behavior, while it may not necessarily hold a mathematical interpretation such as the estimator of the discounted return.

---

> > ### Comment · Reviewer_pvGG · 2023-08-18
> > **Response**
> >
> > Given the response of the authors, I do think this paper is promising and reasonable. Nevertheless, I also think the mathematical interpretation of this method should be improved in future work.

---

### Official Review · Reviewer_C5fY · 2023-07-10

**Soundness:** 2 fair
**Presentation:** 3 good
**Contribution:** 2 fair
**Rating:** 5
**Confidence:** 3

**Summary:**

This paper introduces a reward-free approach termed Proxy Value Propagation (PVP) to facilitate safe and human-aligned reinforcement learning (RL). PVP assigns high Q values to human actions (the state-action pairs in the human demonstration) and low Q values to agent actions that necessitate human intervention. The authors conduct experiments across diverse environments, offering a comparison with existing RL and human-in-the-loop (HL) techniques, as well as the results of user studies.

**Strengths:**

1. This paper is well-written and readable, and provides detailed appendices and codes to supplement the paper's algorithms, experiments, and implementation details.

2. This paper presents a reward-free approach to improve learning from human active involvement, effectively utilizing all data generated during the human intervention, including both agent and human-generated data.

3. The proposed method can be extended to various value-based RL Methods.

**Weaknesses:**

The training process of PVP seems to rely heavily on human involvement, whether through observation or intervention, which can be quite costly, especially when scaling up to more complex scenarios. However, this may not be an issue if the cost is lower than designing safe rewards and the performance is superior to basic RL methods.

Nonetheless, the experimental results and settings in the paper leave me puzzled, and I am unable to determine if the results sufficiently support the conclusions.

1. I am unsure if the experimental setup for Base RL Methods is fair. Why is the Episodic Return so high, but the Success Rate is lower than that of humans? For instance, GT Sophy [1], a race car AI trained using Deep RL by Wurman, Peter R., et al., has already surpassed human champions. Additionally, Imamura, Ryuji, et al. [2] mention that "all agents learn the policy in approximately 400 epochs" and "its score places it among the top 10% approximately 28,000 human players." Can the authors provide more detailed descriptions of the rewards for Base RL Methods? Although the reward settings are briefly described in the appendix, the calculation methods and weights for each reward are not listed. Unreasonable rewards may affect the performance of Base RL Methods and the fairness of comparisons.

2. PVP can be considered as providing dense training signals to the agent, as the agent's training relies on human judgment for every (s, a) pair. Even if the action is not intervened, it still depends on human judgment, thus providing a positive and accurate training signal, leading to faster convergence. For Base RL Methods, can designing more reasonable dense rewards and properly adjusting the weight of crash penalties achieve the same effect? Based on the evaluation metrics, human preference seems to be focused on reducing crash. Can manually designed rewards also express human intentions and preferences? The author should compare the cost of manually crafting the reward with the cost of human intervention, which can highlight the advantages of PVP.

3. I also have doubts about the experimental results for BC and GAIL. What was their training data? Why do human demos have a 97% success rate, while BC and GAIL have success rates of less than 1%? Are there any errors in the experiments?

4. I am also puzzled about the user study results in Table 3. According to the appendix, the scores can only be 1, 2, 3, 4, or 5, with an upper limit of 5 points, and PVP has an average score of 4.8. Why is there a standard deviation of 0.5 when the scores are so high? Are there cases with very low scores, and did the authors analyze these low-score cases in detail? Bad cases can be potential safety hazards.

5. The statement about PVP with reward results is unevidenced: "which might be caused by the fact that the native reward function might not be aligned with human preference." Success rate is a metric of agent capability, so why can't an agent's capability be improved if it is not aligned with human preferences? Is there evidence to prove that RL cannot surpass humans in this task [1, 2]?

6. Additionally, the description of human subjects is insufficient. What is the proficiency of the human subjects in the environments? Were they all novices or professionals? Was a standard test specification or guide provided before the test to ensure consistency in test objectives?

7. Have the authors compared the repetition rates of (s, a) and (s, an) in the Novice Buffer and Human Buffer? This might be closely related to the reliability of human subjects, and learning may not be meaningful for unreliable humans.

8. How much variation is there in (s, an) among different human subjects? When participants have different styles, how should this be handled? For example, if some participants think the current (s, a) is reasonable, but others think it needs intervention, how should this sample be treated?

[1]  Wurman P R, Barrett S, Kawamoto K, et al. Outracing champion Gran Turismo drivers with deep reinforcement learning. Nature, 2022

[2] Imamura R, Seno T, Kawamoto K, et al. Expert Human-Level Driving in Gran Turismo Sport Using Deep Reinforcement Learning with Image-based Representation, 2021

**Questions:**

See the revision recommendations in the "Weaknesses" section. The authors are encouraged to address the aforementioned concerns, provide more detailed experimental information, and incorporate these insights into subsequent revisions.

**Limitations:**

This paper partially discussed the limitations, but there may still be some ethical concerns. It is worth considering whether the proposed method could be potentially abused by malicious human subjects, leading the AI to learn in an incorrect manner.  Can malicious human preferences be detected in a timely manner, and the data be promptly removed?

---

> ### Author Rebuttal · Authors · 2023-08-09
>
> Thank you for your review!
>
> ---
> ### Weaknesses
>
> > **W1:** *I am unsure if the experimental setup for Base RL Methods is fair. Why is the Return so high, but the Success Rate is lower than humans? For instance, GT Sophy has already surpassed human champions. Can the authors provide more detailed descriptions of the rewards?*
>
> **Re:**
>
> First of all, we should emphasize that the essence of this work is to use human online interactions to replace the heavy reward engineering to achieve human-AI alignment.
>
> It is true that through sophisticated reward engineering for a specific task, like what [A] did, the resulting agent can achieve outstanding performance, but it requires extensive domain knowledge and enormous amount of experimentation. Even with a well-defined reward function, the training safety and efficiency of RL remain to be solved.  We demonstrate the success of our method across different tasks with high sample efficiency and safety under human guidance, without even the access to the environment reward in training time.
>
> **The detailed descriptions of the rewards.**
>
> Please refer to the one-page PDF for the detailed formulations of the rewards, which contains four parts: The displacement reward $R_{disp}$, a dense reward to encourage the agent to move toward destination; The speed reward $R_{speed}$; The collision reward $R_{collision}$; and The terminal reward $R_{term}$. This reward function is dense, considers collision penalty, and encourages reaching the destination.
>
> **Why do human subjects have a higher success rate but lower return?**
>
> As you can see from our reward function, high return is possible when the RL agents drive faster but fail before reaching the destination. On the contrary, human subjects might not adopt such high velocity in order to gain more control and make sure the destination is arrived.
>
> **Why can't RL agents surpass human performance in our experiments?**
>
> We do not have a heavily engineered reward function and do not fine tune the weights of different reward terms for different tasks like what [A] did, which is expansive.
>
> On the other hand, imagining we can have a well-engineered reward function for each task that can potentially lead to a super-human policy, the challenges of training-time safety as well as the training efficiency need to be addressed. Our proposed work addresses all these challenges.
>
> ---
>
> > **W2:** *For RL Methods, can designing more reasonable dense rewards and adjusting the weight of crash penalties achieve the human-like behaviors?*
>
> **Re:**
>
> Even though the reward function we used is much simpler than [A], the $R_{disp}$ is a dense reward for task completion and $R_{collision}$ considers the crash penalties and they form a reasonable reward function. It is possible that grid-searching all possible reward weights can lead to better agents. However, such sophisticated reward engineering might need to be performed for each new task that might cost huge computational resources. Meanwhile, the training-time safety and the training efficiency of RL algorithms remain to be solved.
>
> This paper focuses on achieving training-time safety and training efficiency and also the important AI alignment problem, not only on getting a high performing policy through reward engineering.
>
> ---
>
> > **W3:** *The statement about PVP with reward results is unevidenced. Is there evidence to prove that RL cannot surpass humans in this task [A, B]?*
>
> **Re:**
>
> The reward function in [A] consists of 8 terms and the weights of these 8 terms are differently configured for different driving courses (Extended Data Table 1 in [A]). The well-defined reward function and the careful adjustment of reward weights, with other techniques in distributed training, creates the huge success of this Nature paper. Compared to [A], the reward function we used is much simpler and might not be capable of fully capturing human preferences. For example, human subjects usually have more fine grained behaviors such as decelerating before crossroad or maintaining a safe distance from the front vehicles. We believe adopting the task-specific reward function with sufficient weights tuning can let the RL agent surpass humans in our tasks.
>
> We want to point out that, even though we believe reward engineering can finally solve the tasks used in this work and might outperform human players as in [A, B], the training inefficiency and the possible risk during training are still concerning.
>
> In this paper we want to demonstrate the idea that using human active involvement can better communicate human preferences to the learning system while ensuring learning efficiency and safety.
>
> ---
>
> > **W4:** *Doubts about the experimental results for BC and GAIL. Are there any errors in the experiments?*
>
> **Re:**
>
> Dataset size is a critical factor for imitation learning baselines. Table 1 in the submission presents the test performance of the BC and GAIL agents trained by 30K steps collected by human subjects.
>
> To further address your question, we run a CQL baseline on the same 30K steps human dataset. Moreover, we use a trained RL agent as the expert to generate 250K transitions and train baselines on those 250K steps dataset. As shown in the below table, these baselines can achieve decent performance with sufficiently large datasets. Therefore, there are no errors in the implementation in Table 1 and it is the insufficient training data that causes the failure of  BC and GAIL.
>
> |  | Episodic Return | Episodic Cost | Success Rate |
> |---|---|---|---|
> | BC (30K) | 113.3 | 2.17 | 0.07 |
> | GAIL (30K) | 81.5 | 1.30 | 0.0 |
> | CQL (30K) | 156.4 | 1.71 | 0.21 |
> | BC (250K) | 362.1 | 0.13 | 0.57 |
> | GAIL (250K) | 309.6 | 0.68 | 0.60 |
> | CQL (250K) | 373.9 | 0.24 | 0.72 |
>
> ---
>
> [A] Outracing champion Gran Turismo drivers with deep reinforcement learning
>
> [B] Expert Human-Level Driving in Gran Turismo Sport Using Deep Reinforcement Learning with Image-based Representation

---

> > ### Comment · Reviewer_C5fY · 2023-08-17
> >
> > Thank you for your response. However, some of my main concerns still remain unresolved.
> >
> > Q4: **The reliability of the user study results** in Table 3. According to the appendix, the scores can only be 1, 2, 3, 4, or 5, with an upper limit of 5 points, and PVP has an average score of 4.8. Why is there a standard deviation of 0.5 when the scores are so high? I think the results are lacking in plausibility. Could the authors provide more detailed experimental results and a reasonable explanation?
> >
> > Q7&Q8: I have not seen relevant responses yet.

---

> > > ### Author Response · Authors · 2023-08-17
> > > **Response to the follow-up questions**
> > >
> > > Thank you for your follow-up response.
> > >
> > > ---
> > >
> > > > **Q4:** *The reliability of the user study results in Table 3. According to the appendix, the scores can only be 1, 2, 3, 4, or 5, with an upper limit of 5 points, and PVP has an average score of 4.8. Why is there a standard deviation of 0.5 when the scores are so high? I think the results are lacking in plausibility. Could the authors provide more detailed experimental results and a reasonable explanation?*
> > >
> > > **Response:**
> > >
> > > For the item in the user study that has the mean 4.8 and the standard deviation 0.5 (Compliance, and Performance of PVP), the raw scores are typically 4, 5, 5, 5, 5. Given these scores, the average becomes $\frac{4+5+5+5+5}{5} = 4.8$ and standard deviation becomes $\sqrt{ \frac{0.8 ^2 + 0.2 ^ 2 + 0.2^2 + 0.2 ^2 + 0.2^2}{5 - 1} } = 0.447$. We will update the STD to 0.4 due to rounding in Table 3.
> > >
> > >
> > > ---
> > >
> > > > **Q7:** *Have the authors compared the repetition rates of (s, a) and (s, an) in the Novice Buffer and Human Buffer? This might be closely related to the reliability of human subjects, and learning may not be meaningful for unreliable humans.*
> > >
> > > **Response:**
> > >
> > > We haven't compared the repetition rates of $(s, a_n)$ and $(s, a_h)$ in the novice buffer and the human buffer as for continuous state and action space it is hard to evaluate the "repetition".
> > >
> > > As we discussed in the human subject protocol, in the onboarding procedure the human subjects get familiar with the environments and tasks and they can complete several episodes independently before stepping into the main experiments. Therefore we assume the human subjects are not "unreliable".
> > >
> > > ---
> > >
> > > > **Q8:** *How much variation is there in (s, an) among different human subjects? When participants have different styles, how should this be handled? For example, if some participants think the current (s, a) is reasonable, but others think it needs intervention, how should this sample be treated?*
> > >
> > > **Response:**
> > >
> > > For each experiment, there is only one human participant interacting with the system. Therefore there is no cross-participants data stored in the replay buffers when training a novice policy. Policy learning from different human subjects is an important topic for  scaling up the proposed system and is not considered yet in this work.

---

> > > > ### Comment · Reviewer_C5fY · 2023-08-21
> > > >
> > > > Thank you for the further response. I find the problem and method addressed in this paper to be interesting, and I suggest that the authors optimize the user study results in subsequent versions. I will raise my score to accept.
> > > >
> > > > Regarding Q4: Based on the experimental setup in the paper, each participant should have been involved in multiple experiments. However, according to the author's response, there are only 5 ratings. Did each participant provide just one rating? I am unsure when this research questionnaire was completed. If the questionnaire was conducted after all experiments, how could participants identify the algorithms and corresponding performances after numerous experiments? Participants should randomly use different algorithm experiments and then report their preferences for the agent immediately after 1 to 2 episodes of the game. The specific method can be referred to [1, 2]. For methods that rely on human participants, the results of user studies are crucial.
> > > >
> > > > Regarding Q7: Although the participants in the experiments are reliable, measuring human reliability remains very important; otherwise, it may be maliciously exploited by individuals in the application.
> > > >
> > > > [1] Carroll M, et al. On the utility of learning about humans for human-ai coordination. NeurIPS, 2019.
> > > >
> > > > [2] Strouse D J, et al. Collaborating with humans without human data. NeurIPS, 2021.

---

### Official Review · Reviewer_e73G · 2023-07-11

**Soundness:** 4 excellent
**Presentation:** 4 excellent
**Contribution:** 3 good
**Rating:** 8
**Confidence:** 4

**Summary:**

The authors present Proxy Value Propagation for policy optimization. They model a proxy value function so that human intents receive high value (i.e. actions for which the human has inputted as corrective actions over a policy) while on-policy agents have low values for actions that caused an intervention. When optimized via TD-learning they are able to stitch together labeled values of demonstrated state-action pairs with unlabeled data from agent exploration (claimed). They execute the proposed algorithm on a number of environments (e.g. MiniGrid, MetaDrive, CARLA and the Grand Theft Auto 5 driving task) and show significant improvements over baselines.

**Strengths:**

- Human-in-the-loop (with active human involvement) is one of the few tried-and-tested methods to fine tune imitation learning / RL methods on complex real-world domains. Improvements in this field are actually quite impactful to practitioners deploying these systems in production (unlike some other methods that rarely see use outside of the academic setting). To this end the problem is well posed and of interest to the community.

- PVP is a conceptually simple algorithm. I'm pretty sure this paper is self contained and I have everything I need to re implement it.

- The workflow of the algorithm is also practical. I can imagine using this algorithm in a real system with the scale properties such a system would require.

**Weaknesses:**

- nit: "It is also compatible with different forms of human control devices, including gamepad, driving wheel, and keyboard" - True of baselines (and many other methods) as well? Or alternatively, this seems like an obvious point.

- nit: line 128: "It is unrealistic to invite a real human subject to involve in such training." There are many real-world uses of Dagger. Be careful with over generalization.

- I'm not sure the claim that unlabeled (s, a) pairs are stitched together with human expert data is all that well experimentally justified. Or put another way, if the formulation has some equivalency to CQL (with an additional L2 regularization term), in the limit that these regularizes are extremely strong, the method will just look like behavioral cloning. To claim that there is value propagation to unlabeled states I would want to see experiments to confirm this (because in my experience CQL does not actually stitch together sub-optimal trajectories as is often claimed).

- Figure 2 results are quite compelling. Although you're comparing a random-exploration based RL method (and pretty vanilla one at that) against a human-in-the-loop method. This is apples to oranges. I'd also be curious to see what performance looks like against state-of-the-art offline RL or supervised imitation learning in these domains (there's some of this in Table 1, but not in this earlier teaser image).

**Questions:**

- A collection of human demonstrations is assumed (line 143), however it's still not overly clear to me how this is used. Maybe this is already clear in the text but which Q function is actually pre-trained with this data? Presumably there needs to be some overlap between the agent and human preferences initially otherwise every state would induce a human intervention? Maybe spill out this warm up procedure in more detail.

- It's also not entirely clear under what conditions the human is told to intervene? In continuous and high rate action and state spaces (particularly in driving applications), there is not one specific action that causes failure. Are humans instructed to let the agent diverge from an optimal trajectory by a specific metric amount (e.g. half a lane in the case of self driving)? Or is it up to the human operator under what conditions would they intervene? Is the method sensitive to such human ambiguity?

**Limitations:**

First NeurIPS paper where I can comfortably say the limitations section is well done :-) I have nothing else to add.

A societal impact statement doesn't seem necessary for this application / paper.

---

> ### Author Rebuttal · Authors · 2023-08-09
>
> Thank you for your review! Please find our responses as follows.
>
> ---
>
> ### Weaknesses
>
> > **W1:** *nit: "It (the proposed PVP method) is also compatible with different forms of human control devices, including gamepad, driving wheel, and keyboard" - True of baselines (and many other methods) as well? Or alternatively, this seems like an obvious point.*
>
> **Response:**
>
> Different control devices correspond to different action spaces and different human interaction behaviors. For instance, the steering wheel enables a smoother continuous action space compared to the keyboard’s discrete action space by pressing certain keys. These variations pose a generalizability issue for many baseline algorithms.
>
> For example, in Appendix F.1, we show that HACO struggles with the gamepad compared to the steering wheel. On the contrary, PVP demonstrates consistent performance across all tested control interfaces. This highlights PVP's potential for real-world applications where control methods can vary significantly.
>
> ---
>
> > **W2:** *nit: line 128: "It is unrealistic to invite a real human subject to be involved in such training." There are many real-world uses of Dagger. Be careful with overgeneralization.*
>
> **Response:**
>
> Thanks for pointing this out. Real human experiments are expensive, but are not unrealistic. For example, HG-Dagger has been used to train policies in a real car. We will change the wording accordingly, from ‘unrealistic’ to ‘expensive’.
>
> ---
>
> > **W3:** *I'm not sure the claim that unlabeled (s, a) pairs are stitched together with human expert data is all that well experimentally justified. Or put another way, if the formulation has some equivalency to CQL (with an additional L2 regularization term), in the limit that these regularizations are extremely strong, the method will just look like behavioral cloning. To claim that there is value propagation to unlabeled states I would want to see experiments to confirm this (because in my experience CQL does not actually stitch together sub-optimal trajectories as is often claimed).*
>
> **Response:**
>
> Section 5.4 (Table 1 "PVP w/o TD") presents an ablation study showing that removing the TD loss can greatly damage the learning performance. It gives evidence that "there is value propagation to unlabeled states" since if we only use the PVP loss, there will be no value propagation to unlabeled states.
>
> ---
>
> ### Questions
>
> > **Q1:** *A collection of human demonstrations is assumed (line 143), however it's still not overly clear to me how this is used. Maybe this is already clear in the text but which Q function is actually pre-trained with this data? Presumably there needs to be some overlap between the agent and human preferences initially otherwise every state would induce a human intervention? Maybe spill out this warm up procedure in more detail.*
>
> **Response:**
>
> We don’t assume there exist collections of offline human demonstrations beforehand. Therefore, we don't conduct pretraining or warm-up based on held-out human demonstrations.
>
> As clarified in the human subject research protocol (please see the attachment in the common rebuttal), human subjects will provide several episodes of demonstration with full control at the beginning. PVP is efficient enough to learn a rudimentary policy for humans to interact with  after a few minutes of training. This can be seen in the supplementary video.
>
> ---
>
> > **Q2:** *It's also not entirely clear under what conditions the human is told to intervene? In continuous and high rate action and state spaces (particularly in driving applications), there is not one specific action that causes failure. Are humans instructed to let the agent diverge from an optimal trajectory by a specific metric amount (e.g. half a lane in the case of self driving)? Or is it up to the human operator under what conditions would they intervene?*
>
> **Response:**
>
> It is up to the human operator under what conditions they would intervene. We provide a one-page PDF on human subject research protocol in the common rebuttal for more details. We will update Appendix B to include the newly added content.
>
> During the onboarding period, the participants will be briefed about the objectives of all driving tasks: to safely maneuver the vehicle to its designated destination while ensuring the vehicle's operation aligns with traffic regulations and human preferences. The participants will practice each task with different control devices.
>
> The human subjects will perform intervention whenever they perceive the vehicle might be in a dangerous situation, in violation of traffic rules, or in whatever scenario they feel they wouldn't behave in the way the novice policies do.

---

> > ### Comment · Area_Chair_D8zF · 2023-08-19
> >
> > Dear authors,
> >
> > Thank you for your taking the time to respond to the comments.
> >
> > Dear Reviewer e73G,
> >
> > After reading the authors' response, do you have any additional thoughts?
> >
> > Best,
> >
> > AC

---

### Author Rebuttal · Authors · 2023-08-09

In this common rebuttal, we provide a one-page PDF containing two sections:

1. Human subject research protocol: including information on human subjects recruitment, onboarding procedure, the information we provide to them, the main experiment and the questionnaire. We will update Appendix B to include newly added content.
2. Reward function for driving tasks: describing the detailed reward function we used. It contains four terms, a dense displacement reward which measures the progress toward destination, a speed reward, a collision penalty and a termination reward.

---

> ### Comment · Reviewer_sDoy · 2023-08-11
> **Possible typo in the reward function**
>
> It seems the collision term of the reward function is positive reward with a positive multiple (+1 for collision, multiple of 5). Is one of these terms supposed to be negative (either the reward or the scaling factor)? Otherwise it seems the car would be encouraged to collide with obstacles.

---

> > ### Author Response · Authors · 2023-08-12
> >
> > Thank you for pointing out the typo.
> >
> > When training RL baselines such as SAC, TD3 and PPO. The collision penalty is actually $-5$ when a collision happens. At this moment the episode is also terminated.
> >
> > When training the safe RL benchmark, e.g. PPO-Lagrangian, and evaluating all trained policies from any algorithm, the collision penalty is set to $-1$ per collision and the episode will not be terminated due to collision. That's how the episodic safety cost is computed in Table 1.

---

### Decision · Program_Chairs · 2023-09-21

**Decision:**

Accept (spotlight)

**Comment:**

The paper introduces PVP that leverages human intervention as a signal of action quality instead of relying on ground-truth rewards. PVP assigns high Q values to actions taken by humans and low Q values to actions requiring human corrections. By relabeling human interventions with positive Q value targets and penalizing the agent's erroneous actions, the policy is trained through offline reinforcement learning.


**strengths**

* Extensive comparisons with prior work and multiple baselines are conducted in the experiments.
* Human experiments that ensure real-world applicability and avoid synthetic/heuristic labeling.
* The proposed method is presented in a clear and intuitive manner.
* The paper's approach is noted for its elegance and simplicity, making it potentially applicable to various domains.

**weaknesses/suggestions**

* It would be nice if the authors could clarify the details of the user study in the final version.
* Current experiments are limited to navigation environments. It would strengthen the paper to show the results on robot manipulation tasks.

In summary, the paper presents the PVP method as a compelling approach to reinforcement learning with human involvement, backed by comprehensive experiments and clear presentation, making it an impactful contribution to the field.